# Discovering and Explaining the Representation Bottleneck of DNNs

**Huiqi Deng,**[*] **Qihan Ren,**[*] **Hao Zhang, Quanshi Zhang**[†]
Shanghai Jiao Tong University
`{denghq7,renqihan,1603023-zh,zqs1022}@sjtu.edu.cn`

## Abstract

This paper explores the bottleneck of feature representations of deep neural networks (DNNs), from the perspective of the complexity of interactions between input variables encoded in DNNs. To this end, we focus on the multi-order interaction between input variables, where the order represents the complexity of interactions. We discover that a DNN is more likely to encode both too simple and too complex interactions, but usually fails to learn interactions of intermediate complexity. Such a phenomenon is widely shared by different DNNs for different tasks. This phenomenon indicates a cognition gap between DNNs and humans, and we call it a *representation bottleneck*. We theoretically prove the underlying reason for the representation bottleneck. Furthermore, we propose losses to encourage/penalize the learning of interactions of specific complexities, and analyze the representation capacities of interactions of different complexities. The code is available at `https://github.com/Nebularaid2000/bottleneck`.

## 1 Introduction

The revolution from shallow to deep models is a crucial step in the development of artificial intelligence. DNNs usually exhibit superior performance to shallow models, which is generally believed as a result of the improvement of the representation power (Pascanu et al., 2013; Montúfar et al., 2014). To this end, instead of considering previous issues of the accuracy and the generalization ability of DNNs, we focus on the following two questions about the representation capacity:
• Are there any common tendencies of DNNs in representing specific types of features?
• Does a DNN encode similar visual concepts as human beings for image classification?

In order to answer the above two questions, we first investigate the bottleneck of feature representation, *i.e.*, which types of concepts are likely to be encoded by a DNN, and which types of concepts are difficult to be learned. To this end, we discover that the interaction between input variables is an effective tool to analyze the feature representation. It is because instead of considering input variables working independently, the DNN encodes the interaction between input variables to form an interaction pattern for inference. For example, the inference of a face image can be explained as the interactions between left and right eyes, between nose and mouth, etc.

As the answers to the above questions, we discover a common representation bottleneck of DNNs in encoding interactions, *i.e.*, a DNN is more likely to encode both too complex and too simple interactions, instead of encoding interactions of intermediate complexity. This bottleneck also indicates a dramatic difference between the inferences of DNNs and humans.

The interaction can be understood as follows. Let us take the face recognition task for example. Let $\phi_{i=\text{mouth}}$ measure the numerical importance of the mouth region $i$ to the classification score. Then, the interaction utility between the mouth region $i$ and the nose region $j$ is measured as the change of the $\phi_{i=\text{mouth}}$ value by the presence or absence of the nose region $j$. If the presence of $j$ increases the importance $\phi_{i=\text{mouth}}$ by 0.1, then, we consider 0.1 as the utility of the interaction between $i$ and $j$.

---

[*]Equal contribution.
[†]Quanshi Zhang is the corresponding author. He is with the John Hopcroft Center and the MOE Key Lab of Artifical Intelligence, AI Institute, at the Shanghai Jiao Tong University, China.

(a) Interaction pattern (b) Representation bottleneck (c) Whether humans/DNNs extract new information from patches.

Figure 1: (a) Five pixels $(g, h, i, j, k)$ interact with each other, forming an edge pattern for classification. (b) Representation bottleneck. A DNN is likely to encode low-order and high-order interactions, but usually fails to learn middle-order interactions. (c) The cognition gap between DNNs and humans. Humans extract little information from a few image patches (*e.g.*, $5\%$ patches). Also, given almost all patches (*e.g.*, $90\%$ patches), people learn little new information from the additional $5\%$ patches since the information is already redundant for human recognition. In comparison, the DNN encodes most interactions when the DNN is given very few patches or most patches.

**Multi-order interactions.** In order to represent the interaction complexity mentioned in the representation bottleneck, we use the multi-order interaction utility between variables $i, j$ proposed by Zhang et al. (2020). The interaction of the $m$-th order $I^{(m)}(i, j)$ measures the average interaction utility between variables $i, j$ on all contexts consisting of $m$ variables. In this way, the order $m$ reflects the contextual complexity of the interaction. A low-order $I^{(m)}(i, j)$ measures the relatively simple collaboration between variables $i, j$, and a few $m$ contextual variables, while a high-order $I^{(m')}(i, j)$ corresponds to the complex collaboration between $i, j$ and $m'$ massive contextual variables, where $m' \gg m$.

Moreover, we prove that the multi-order interaction is a trustworthy tool to analyze the representation capacity of DNNs. Specifically, the output score of a DNN can be decomposed into utilities of compositional multi-order interactions between different pairs of variables, *i.e.*, *model output* $= \sum_{m=0}^{n-2} \sum_{i,j \in N, i \neq j} w^{(m)} I^{(m)}(i, j) + \sum_{i \in N}$ *local utility of $i$ + bias*. For example, the inference score of a face can be decomposed into the interaction utility between left and right eyes, between mouth and nose, etc. Therefore, we can take the utility $I^{(m)}(i, j)$ as the underlying reason to explain the DNN, because each interaction makes a compositional contribution to the output.

**Representation bottleneck of DNNs.** Surprisingly, the above decomposition of multi-order interactions enables us to discover a representation bottleneck of DNNs. As Figure 1(b) shows, low-order and high-order interaction utilities $I^{(m)}(i, j)$ usually have high absolute values, while middle-order interaction utilities $I^{(m)}(i, j)$ usually have low absolute values. In other words, a DNN is more likely to encode the interaction between variables $i$ and $j$, when $i, j$ interact with a few contextual variables. Similarly, it is also easy for the DNN to learn the interaction, when $i, j$ interact with most contextual variables. However, it is difficult for the DNN to learn the interaction, when $i, j$ cooperate with a medium number of contextual variables. The difficulty of learning middle-order interactions reflects a representation bottleneck of DNNs.

**Cognitive gap between DNNs and humans.** Such a representation bottleneck also indicates a significant gap between the concepts encoded by DNNs and the visual cognition of humans. As Figure 1(c) shows, people usually cannot extract meaningful information from a few image patches. Besides, if people are given almost all patches, then the information is already too redundant and inserting additional patches will bring in little new information. In contrast, the DNN encodes most information, when the DNN is given only a few patches or is given most patches.

**Theoretical proof.** In this paper, we theoretically prove the mechanism that is responsible for the representation bottleneck. Such proof also enables us to simulate the distribution of interactions of different orders, which well matches the distribution of interactions in real applications.

Beyond the theoretical proof, another important issue is how to guide the learning of feature representation in DNNs by learning interactions of specific orders. We propose two losses to encourage/penalize the DNN to make inferences by interactions of specific orders, thereby boosting/preventing the learning of such interactions. Experimental results have validated the effectiveness of the two losses. Next, we investigate the representation capacities of several DNNs which encoded interactions of different orders. We find that the DNNs mainly encoding high-order interactions represent more structural information than the normally trained DNNs. In addition, high-order interactions were vulnerable to adversarial attacks.

In summary, this paper makes three contributions:

• This study discovers a representation bottleneck phenomenon of DNNs, *i.e.*, it is difficult for a DNN to learn middle-order interactions. It also clearly proves that DNNs and humans use different types of visual concepts for inference.

• We theoretically prove the underlying reason for the representation bottleneck.

• We design two losses to encourage/penalize the DNN to learn interactions of specific orders. Experiments have validated the effectiveness of the proposed losses. Besides, we investigate the representation capacities of DNNs which encode interactions of different orders.

## 2 RELATED WORK

**The representation capacity of DNNs.** The evaluation of the representation capacity of DNNs provides a new perspective to explain and analyze DNNs. Pascanu et al. (2013) and Montúfar et al. (2014) used the number of linear response regions in a deep rectifier MLP to evaluate its representation capacity. The information bottleneck theory (Shwartz-Ziv & Tishby, 2017) used the mutual information to explain how DNNs gradually learned the information during the training process. Achille & Soatto (2018), Amjad & Geiger (2019) and Hjelm et al. (2019) further improved the representation capacity by optimizing mutual information. Arpit et al. (2017) studied the memorization behavior of DNNs during training to analyze the feature representations. Xu (2018) proposed Fourier analysis to understand the generalization. In addition, several metrics were proposed to analyze the generalization capacity or robustness of DNNs, including the stiffness (Fort et al., 2019), the sensitivity (Novak et al., 2018), and the CLEVER score (Weng et al., 2018). Neyshabur et al. (2017) examined whether existing complexity measures can guarantee generalization.

Previous researches mainly studied the theoretical maximum complexity, generalization ability, and robustness of DNNs. In comparison, our research focuses on the limitation of DNNs in feature representations, *i.e.*, which types of interactions are unlikely to be encoded.

**Interactions.** Interactions between input variables of a DNN have been widely investigated in recent years. Based on the Shapley value (Shapley, 1951), Grabisch & Roubens (1999) proposed the Shapley interaction index to define the interaction in a cooperative game. Lundberg et al. (2018) used the interaction to build tree ensemble explanations for DNNs. Janizek et al. (2021) extended Integrated Gradients (Sundararajan et al., 2017) to explain the pairwise feature interaction in DNNs. Sundararajan et al. (2020) proposed the Shapley Taylor interaction to measure interactions among multiple variables. Tsang et al. (2020) and Tsang et al. (2017) interpreted DNNs by detecting statistical interactions between input variables and interactions between network weights, respectively. Peebles et al. (2020) and Tsang et al. (2018) achieved the disentanglement of features by restricting interactions. Song et al. (2019) and Lian et al. (2018) designed network architectures to effectively learn feature interactions. Lengerich et al. (2020) applied the ANOVA technique to measure interactions and further explored the relationship between dropout and interactions. Zhang et al. (2020) proposed the multi-order interaction, and used it to understand and boost dropout.

Besides, the team of Dr. Quanshi Zhang has adopted the game-theoretic interactions to build up a theoretical system to explain the representation capacity of a DNN, including explaining the generalization ability (Zhang et al., 2020), the adversarial transferability and adversarial attacks (Wang et al., 2021b;a) of a DNN, and explaining concepts encoded in a DNN (Cheng et al., 2021; Ren et al., 2021a; Zhang et al., 2021b;a).

## 3 REPRESENTATION BOTTLENECK

Before the analysis of the representation bottleneck, let us first introduce multi-order interactions between input variables, which are encoded in a DNN. Given a pre-trained DNN $v$ and an input sample with a set of $n$ variables $N = \{1, \ldots, n\}$ (*e.g.*, an input image with $n$ pixels), $v(N)$ denotes the network output of all input variables. Input variables of DNNs usually interact with each other to make inferences, instead of working individually. In this study, we mainly discuss the pairwise interactions. For example, as Figure 1(a) shows, pixels $i, j \in N$ interact with each other, forming an edge pattern for classification. If the existence of this pattern increases the network output by 0.01, we consider this pattern has a positive *utility* of 0.01. Similarly, if the existence of this pattern decreases the network output, we consider this pattern has a negative utility.

Furthermore, the multi-order interaction $I^{(m)}(i,j)$ between two input variables $i, j \in N$, $0 \le m \le n-2$, was proposed to measure interactions of different complexities (Zhang et al., 2020). Specifically, the $m$-th order interaction $I^{(m)}(i,j)$ measures the average interaction utility between variables $i, j$ under all possible contexts consisting of $m$ variables. Therefore, the order $m$ can be considered to represent the contextual complexity of the interaction. For example, as Figure 1(a) shows, five pixels $(i, j, g, h, k)$ collaborate with each other and form an edge pattern for classification. Thus, the pairwise interaction between pixels $i, j$ also depends on the three contextual pixels $g, h, k$. Mathematically, the multi-order interaction $I^{(m)}(i,j)$ is defined as follows:

$$I^{(m)}(i,j) = \mathbb{E}_{S \subseteq N \setminus \{i,j\}, |S|=m}[\Delta v(i,j,S)], \tag{1}$$

where $\Delta v(i,j,S) = v(S \cup \{i,j\}) - v(S \cup \{i\}) - v(S \cup \{j\}) + v(S)$. Here, $v(S)$ is the output score when we keep variables in $S \subseteq N$ unchanged but replace variables in $N \setminus S$ by the baseline value. The baseline value follows the widely-used setting in Ancona et al. (2019), which is set as the average value of the variable over different samples. Let us take the multi-category image classification for example. Given an input image $x$, $v(S) = v(x_S)$ can be implemented as any scalar output of the DNN (*e.g.*, $\log \frac{P(\hat{y}=y^{\text{truth}}|x_S)}{1-P(\hat{y}=y^{\text{truth}}|x_S)}$ of the true category), where we replace the pixel values in $N \setminus S$ of original input $x$ by the baseline value (the average pixel value over images) to construct a masked image $x_S$. Then, $\Delta v(i,j,S) = [v(S \cup \{i,j\}) - v(S \cup \{i\})] - [v(S \cup \{j\}) - v(S)]$ quantifies the marginal effects (the importance) of the variable $j$ that are changed by the presence or absence of the variable $i$. It represents the utilities of the collaboration between $i, j$ in a context $S$.

**Generic metric.** The proposed multi-order interaction $I^{(m)}(i,j)$ is a generic metric, which has a strong connection with the Shapley value (Shapley, 1951) and the Shapley interaction index (Sundararajan et al., 2020) in game theory. In addition, it has been proven that $I^{(m)}(i,j)$ satisfies the following five desirable properties, *i.e.*, *linearity*, *nullity*, *commutativity*, *symmetry*, and *efficiency* properties. The connections with existing metrics and five properties are introduced in Appendix A.

## 3.1 REPRESENTATION BOTTLENECK

According to the ***efficiency*** property of $I^{(m)}(i,j)$, we find that the output of a DNN can be explained as the sum of all interaction utilities of different orders between different pairs of variables.

$$v(N) = v(\emptyset) + \sum_{i \in N} \mu_i + \sum_{i,j \in N, i \neq j} \sum_{m=0}^{n-2} w^{(m)} I^{(m)}(i,j) \tag{2}$$

where $\mu_i = v(\{i\}) - v(\emptyset)$, and $w^{(m)} = (n-1-m)/[n(n-1)]$. Because $I^{(m)}(i,j)$ measures the interaction between variables $i$ and $j$ encoded in DNNs with $m$ contextual variables, we can consider the interaction utility $I^{(m)}(i,j)$ as a specific reason for the inference, which makes a compositional contribution $w^{(m)} I^{(m)}(i,j)$ to the output.

In this way, we can categorize all underlying reasons for the network output into different complexities. Low-order interactions can be considered as simple underlying reasons, relying on very few variables. High-order interactions can be regarded as complex underlying reasons, depending on massive variables. In order to measure the reasoning complexity of the DNN, we measure the relative interaction strength $J^{(m)}$ of the encoded $m$-th order interaction as follows:

$$J^{(m)} = \frac{\mathbb{E}_{x \in \Omega}[\mathbb{E}_{i,j}[|I^{(m)}(i,j|x)|]]}{\mathbb{E}_{m'}[\mathbb{E}_{x \in \Omega}[\mathbb{E}_{i,j}[|I^{(m')}(i,j|x)|]]]} \tag{3}$$

where $\Omega$ denotes the set of all samples. $J^{(m)}$ is computed over all pairs of input variables in all samples. $J^{(m)}$ is normalized by the average value of interaction strength. The distribution of $J^{(m)}$ measures the distribution of the complexity of interactions encoded in DNNs.

**Representation bottleneck.** Based on the above metric, we discover an interesting phenomenon: *a DNN usually encodes strong low-order and high-order interactions, but encodes weak middle-order interactions*. Such a phenomenon is shared by different DNN architectures trained on different datasets, which is illustrated by the $J^{(m)}$ curves in Figure 2. Specifically, when the order $m$ is smaller than $0.1n$ or greater than $0.9n$, the interaction strength $J^{(m)}$ is usually high. In comparison, $J^{(m)}$ is usually low when the order $m$ approximates $0.5n$. Moreover, Figure 3(a) shows that such a phenomenon does not only exists in well-trained DNNs, but also exists in the entire training process.

The above phenomenon indicates that a DNN is more likely to learn simple interactions where a few variables (*e.g.*, less than $0.1n$ variables) interact with each other. Similarly, it is easy for a DNN to

Figure 2: The distributions of interaction strength $J^{(m)}$ of different DNNs trained on various image datasets and tabular datasets.[2]

encode complex interactions where massive variables (*e.g.*, more than $0.9n$ variables) participate. However, it is difficult for a DNN to learn middle-complex interactions in which a medium number of variables (*e.g.*, about $0.5n$ variables) participate. Let us take Figure 1(c) for an example. When a DNN is given very few patches sparsely distributed on the horse image, the DNN can successfully extract the interaction between the few patches. Similarly, when the DNN is given almost all patches, then the insertion of two new patches will make the DNN trigger strong collaboration between the two patches and massive existing patches. However, when the DNN is just given a half patches, it is difficult for the DNN to encode interactions between the two patches. In a word, the encoded interaction pattern is either too simple or too complex. The difficulty of learning interaction patterns of moderate complexity reflects a common tendency in the feature representation of DNNs.

Such a representation bottleneck also indicates that DNNs and human beings encode different types of visual patterns for inference. As Figure 1(c) shows, (i) Given very few patches, a DNN can encode much information from low-order interaction patterns between patches. However, it is difficult for people to recognize such low-order interactions. (ii) Given almost all patches of an image, any additional patches are already too redundant for human cognition, so people do not obtain much new information from additional patches. (iii) Given a medium number of patches, the DNN usually extracts little information, while people can extract much information for recognition.

**Implementation details**. In order to measure $J^{(m)}$, we conducted experiments on three image datasets including the ImageNet dataset (Russakovsky et al., 2015), the Tiny-ImageNet dataset (Le & Yang, 2015) and the CIFAR-10 dataset (Krizhevsky et al., 2009). We mainly analyzed several DNNs trained on these datasets for image classification, including AlexNet (Krizhevsky et al., 2012), VGG-16 (Simonyan & Zisserman, 2014) and ResNet-18/20/50/56 (He et al., 2016). Due to the high dimension of input variables ($n = 224 \times 224$ for ImageNet), the computational cost of $J^{(m)}$ is intolerable. To reduce the computational cost, we split the input image into $16 \times 16$ patches, and considered each patch as an input variable. To compute $J^{(m)}$, we set $v(S|x) = \log \frac{P(\hat{y}=y^*|x_S)}{1-P(\hat{y}=y^*|x_S)}$ given the masked sample $x_S$, where $y^*$ is the true label and $P(\hat{y} = y^*|x_S)$ is the probability of classifying the masked sample $x_S$ to the true category. In the masked sample $x_S$, pixel values in image patches in $N \setminus S$ were replaced by the average pixel value over different patches in all images, just like in Ancona et al. (2019). Note that $J^{(m)}$ is an average over all possible contexts $S$, all pairs of variables $(i, j)$, and all samples $x$, which is computationally infeasible. Therefore, we approximated $J^{(m)}$ using a sampling strategy (Zhang et al., 2020). Please see Appendix C for sampling details. In addition, we conducted experiments on two tabular datasets, including the UCI census income dataset (census) and the UCI TV news channel commercial detection dataset (commercial) (Dua et al., 2017). Each sample in the two datasets contained $n = 12$ and $n = 10$ input variables, respectively. We analyzed a five-layer MLP (namely, MLP-5) and an eight-layer MLP (namely, MLP-8) network. Each layer except for the output layer contained 100 neurons. In the computation of $J^{(m)}$, $P(\hat{y} = y^*|x_S)$ was also computed by setting the baseline value of variable to the average value of the variable. Please see Appendix C for details.

## 3.2 EXPLAINING THE REPRESENTATION BOTTLENECK

In this subsection, we theoretically prove the underlying reason for the representation bottleneck. Let $W \in \mathbb{R}^K$ denote the network parameters of a DNN. We focus on the change $\Delta W$ of network parameters, which also represents the strength of training the DNN. The change of weights is calcu-

---

[2]Note that the strength of high-order interactions on tabular datasets is higher than that on image datasets. It may be because the image classification usually depends on low-order interactions, but the classification on tabular data mainly depends on high-order interactions, thereby forcing DNNs to learn high-order interactions.

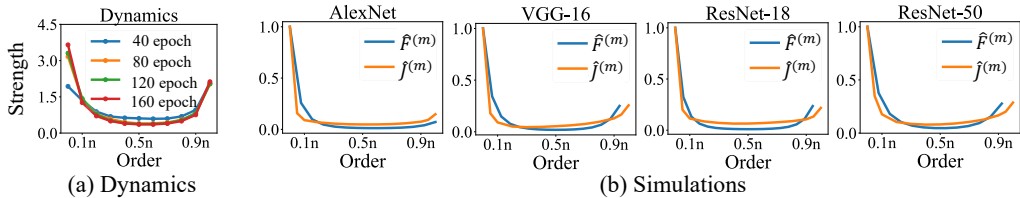

Figure 3: (a) Distributions of the interaction strength $J^{(m)}$ of a ResNet-20 model over different orders, which were measured after different training epochs. The DNN was trained on the CIFAR-10 dataset. (b) Simulations of the $\hat{J}^{(m)}$ distributions based on $\hat{F}^{(m)}$ curves on the ImageNet dataset.

lated by $\Delta W = -\eta \frac{\partial L}{\partial W} = -\eta \frac{\partial L}{\partial v(N)} \frac{\partial v(N)}{\partial W}$. Here, $L$ denotes the loss function, and $\eta$ is the learning rate. According to Eq. (2), the network output $v(N)$ of the DNN can be decomposed into the sum of multi-order interactions $I^{(m)}(i,j)$. Therefore, $\Delta W$ can be further represented as the sum of gradients $\frac{\partial I^{(m)}(i,j)}{\partial W}$ of multi-order interactions.

$$\Delta W = -\eta \frac{\partial L}{\partial v(N)} \frac{\partial v(N)}{\partial W} = \Delta W_U + \sum_{m=0}^{n-2} \sum_{i,j \in N, i \neq j} \Delta W^{(m)}(i,j), \qquad (4)$$

where $U = v(\emptyset) + \sum_{i \in N} \mu_i$. Specifically,

$$\Delta W_U \overset{\text{def}}{=} -\eta \frac{\partial L}{\partial v(N)} \frac{\partial v(N)}{\partial U} \frac{\partial U}{\partial W}, \quad \Delta W^{(m)}(i,j) \overset{\text{def}}{=} R^{(m)} \frac{\partial I^{(m)}(i,j)}{\partial W},$$

where $R^{(m)} = -\eta \frac{\partial L}{\partial v(N)} \frac{\partial v(N)}{\partial I^{(m)}(i,j)}$. Here, $\Delta W_U$ represents the component of $\Delta W$ w.r.t. $\frac{\partial U}{\partial W}$, and $\Delta W^{(m)}(i,j)$ represents the component of $\Delta W$ w.r.t. $\frac{\partial I^{(m)}(i,j)}{\partial W}$. Therefore, besides $\Delta W_U$, we can consider there are additional $\frac{n(n-1)^2}{2}$ paths w.r.t. different pairs of $i,j$ and different orders $m$ in the backpropagation, and the weight change through each propagation path is $\Delta W^{(m)}(i,j)$. In this way, we can consider the norm of $\Delta W^{(m)}(i,j)$ (i.e., $||\Delta W^{(m)}(i,j)||_2$) measures the strength of learning the interaction between variables $i$ and $j$ under contexts of $m$ variables.

**Theorem 1.** (*Proof in Appendix B*) Assume $\mathbb{E}_{i,j,S}[\frac{\partial \Delta v(i,j,S)}{\partial W}] = \mathbf{0}$. Let $\sigma^2$ denote the variance of each dimension of $\frac{\partial \Delta v(i,j,S)}{\partial W}$. Then, $\mathbb{E}_{i,j}[\Delta W^{(m)}(i,j)] = \mathbf{0}$ and the variance of each dimension of $\Delta W^{(m)}(i,j)$ is $(\eta \frac{\partial L}{\partial v(N)} \frac{n-m-1}{n(n-1)})^2 \sigma^2 / \binom{n-2}{m}$. Therefore, $\mathbb{E}_{i,j}[||\Delta W^{(m)}(i,j)||_2^2] = K(\eta \frac{\partial L}{\partial v(N)} \frac{n-m-1}{n(n-1)})^2 \sigma^2 / \binom{n-2}{m}$, where $K$ is the dimension of the network parameter $W$.

Theorem 1 shows that the strength (*i.e.*, the $l_2$-norm $||\Delta W^{(m)}(i,j)||_2$) of learning $m$-order interactions is proportional to $F^{(m)} = \frac{n-m-1}{n(n-1)}/\sqrt{\binom{n-2}{m}}$. Therefore, when the order $m$ is small or large (*e.g.*, $m = 0.05n$ or $0.95n$), the training strength of the $m$-order interaction is relatively higher. In contrast, when the order $m$ is medium (*e.g.*, $m = 0.5n$), the training strength of the $m$-order interaction is much lower. The above analysis explains why it is easy for a DNN to learn low-order and high-order interactions, but difficult for a DNN to learn middle-order interactions.

**Simulation of the curve of the interaction strength.** We find that the above training strength can be used to simulate the distribution of interaction strengths $J^{(m)}$ in real applications, which verifies our theory. Based on Theorem 1, the training strength w.r.t. the order $m$ is proportional to the aforementioned $F^{(m)}$, so we can use $F^{(m)}$ to simulate $J^{(m)}$. For fair comparison, we normalized $F^{(m)}$ and $J^{(m)}$ by $\hat{F}^{(m)} = F^{(m)}/F^{(0)}$ and $\hat{J}^{(m)} = J^{(m)}/J^{(0)}$, such that $\hat{F}^{(0)} = \hat{J}^{(0)} = 1$. Figure 3(b) shows that the curves of $\hat{F}^{(m)}$ can well match the distributions of $\hat{J}^{(m)}$. Due to the redundancy of feature representations in DNNs, we usually consider that the actual dimension $n'$ of the latent space of DNNs is much lower than the number $n$ of input variables. Thus, instead of directly using the number $n$ of input variables, we adopted a smaller $n'$ (*i.e.*, $n' < n$) in $\hat{F}^{(m)}$ for the simulation.

### 3.3 METHOD TO CONTROL INTERACTIONS OF SPECIFIC ORDERS

The representation bottleneck is widely shared by DNNs of different architectures for various tasks, when these DNNs are normally trained. In this section, we mainly explore methods, which force the

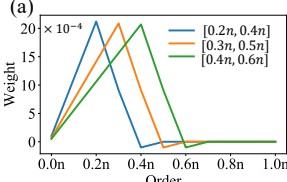 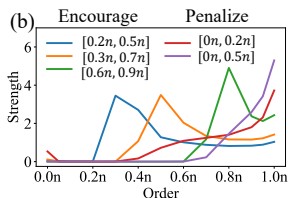 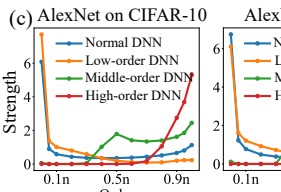

Figure 4: (a) Weight coefficients $\tilde{w}^{(m)}$ of different orders with different pairs of $(r_1, r_2)$. (b) Distributions of the interaction strength $J^{(m)}$ over different orders. Each curve indicates a AlexNet whose interactions were encouraged/penalized by $L^+(r_1, r_2)$ and $L^-(r_1, r_2)$ with certain pairs of $(r_1, r_2)$. (c) Distributions of $J^{(m)}$ of four types of DNNs (AlexNet). The Appendix C provides more results.

DNN to learn interactions of specific orders. In this way, we can investigate the properties of feature representations of such DNNs.

In order to force the DNN to learn interactions of specific orders, we propose two simple-yet-efficient losses in the training process. The two losses encourage and penalize interactions of specific orders, respectively. Before designing the two losses, let us focus on the output change $\Delta u(r_1, r_2)$:

$$\Delta u(r_1, r_2) = \mathbb{E}_{S_1, S_2 : \emptyset \subseteq S_1 \subsetneq S_2 \subseteq N}[v(S_2) - r_2/r_1 \cdot v(S_1)], \tag{5}$$

where the subsets $S_1$ and $S_2$ are randomly sampled from all input variables $N$, such that $\emptyset \subseteq S_1 \subsetneq S_2 \subseteq N$, $|S_1| = r_1 n$, $|S_2| = r_2 n$, and $0 \leq r_1 < r_2 \leq 1$.

**Theorem 2.** (*Proof in Appendix B*) The output change $\Delta u(r_1, r_2)$ can be decomposed into the sum of multi-order interactions between different pairs of variables.

$$\Delta u(r_1, r_2) = (1 - r_2/r_1)v(\emptyset) + \sum_{m=0}^{n-2} \sum_{i,j \in N, i \neq j} \tilde{w}^{(m)} I^{(m)}(i, j)$$

$$\text{where} \quad \tilde{w}^{(m)} = \begin{cases} (r_2/r_1 - 1)(m+1)/[n(n-1)], & m \leq r_1 n - 2 \\ (r_2 n - m - 1)/[n(n-1)], & r_1 n - 2 < m \leq r_2 n - 2 \\ 0, & r_2 n - 2 < m \leq n - 2 \end{cases} \tag{6}$$

Interestingly, as Figure 4(a) shows, we can consider that the output change $\Delta u(r_1, r_2)$ mainly encodes interactions whose orders are in the range of $[0, r_2 n]$. The weight coefficient $\tilde{w}^{(m)}$ of the $m$-th order interaction reaches a peak at the $r_1 n$-th order in $\Delta u(r_1, r_2)$.

**Encourage/penalize interactions of specific orders.** Based on the above analysis, $\Delta u(r_1, r_2)$ only contains partial interactions of the $[0, r_2 n]$-th orders. Hence, we propose two losses based on $\Delta u(r_1, r_2)$, which encourage and penalize the DNN to use interactions of specific orders for inference, respectively. The first proposed loss $L^+(r_1, r_2)$ forces the DNN to mainly use interactions encoded in $\Delta u(r_1, r_2)$ for inference, thereby boosting the learning of these interactions.

$$L^+(r_1, r_2) = -\frac{1}{|\Omega|} \sum_{x \in \Omega} \sum_{c=1}^{C} P(y^* = c|x) \log P(\hat{y} = c|\Delta u_c(r_1, r_2|x)), \tag{7}$$

where $L^+(r_1, r_2)$ is the cross entropy that uses $\Delta u(r_1, r_2)$ for classification. Here, $\Omega$ is the training set, and $C$ denotes the number of classes. Given an input image $x \in \Omega$, $y^*$ is the true label, and $\hat{y}$ denotes the predicted label. Here, $\Delta u_c(r_1, r_2|x) = v_c(S_2|x) - r_2/r_1 \cdot v_c(S_1|x)$ denotes the change of the logits of the category $c$, where the logit $v_c(S|x)$ denotes the feature dimension corresponding to the $c$-th category before the softmax layer. The two subsets $S_1, S_2$ are randomly sampled. In this way, we compute $P(\hat{y} = c|\Delta u_c(r_1, r_2|x))$ as the probability of using the $C$-dimensional vector $\{\Delta u_c(r_1, r_2|x)|1 \leq c \leq C\}$ to classify the sample to the category $c$. We input the $C$-dimensional vector $\{\Delta u_c(r_1, r_2|x)|1 \leq c \leq C\}$ into the softmax layer to compute the probability.

Besides, the second loss $L^-(r_1, r_2)$ is designed to prevent the DNN from encoding interactions of the $[0, r_2 n]$-th orders. Specifically, we maximize the entropy of classification based on $\Delta u(r_1, r_2)$, in order to make $\Delta u(r_1, r_2)$ non-discriminative.

$$L^-(r_1, r_2) = \frac{1}{|\Omega|} \sum_{x \in \Omega} \sum_{c=1}^{C} P(\hat{y} = c|\Delta u_c(r_1, r_2|x)) \log P(\hat{y} = c|\Delta u_c(r_1, r_2|x)), \tag{8}$$

where $L^-(r_1, r_2)$ denotes the minus entropy of the classification probability based on $\Delta u_c(r_1, r_2)$.

In this way, we can train a DNN using the following loss,

$$\text{Loss} = \text{Loss}_{\text{classification}} + \lambda_1 L^+(r_1, r_2) + \lambda_2 L^-(r_1, r_2) \tag{9}$$

Table 1: (left) Classification accuracies of four types of DNNs, including the normally trained DNNs, and the other three types of DNNs mainly encoding low-order, middle-order, and high-order interactions. (right) Comparison of adversarial accuracies between normally trained DNNs and DNNs mainly encoding high-order interactions on the census dataset and the commercial dataset.

| Model | CIFAR-10 | | | Tiny-ImageNet | | | Model | Normal training | Penalize low-order & boost high-order |
|---|---|---|---|---|---|---|---|---|---|
| | AlexNet | VGG16 | VGG19 | AlexNet | VGG16 | VGG19 | | | |
| Normal training | 88.52 | 90.50 | 90.61 | 56.00 | 56.16 | 52.56 | MLP-5 on census | 38.22 | **7.31** |
| Low interaction | 86.97 | 89.99 | 89.74 | 58.68 | 55.60 | 55.04 | MLP-8 on census | 39.33 | **2.02** |
| Mid interaction | 86.65 | 90.29 | 90.03 | 53.88 | 55.84 | 53.36 | MLP-5 on commer | 27.01 | **22.00** |
| High interaction | 88.68 | 90.84 | 90.79 | 56.12 | 55.36 | 53.28 | MLP-8 on commer | 25.92 | **20.58** |

where $\lambda_1 \geq 0, \lambda_2 \geq 0$ are two constants to balance the three terms.

**Effects of the two losses.** In experiments, we found that the loss $L^-(r_1, r_2)$ usually could successfully penalize interactions of the $[r_1 n, r_2 n]$-th orders and $L^+(r_1, r_2)$ could encourage interactions of the $[r_1 n, r_2 n]$-th orders, instead of penalizing/encouraging interactions of the $[0, r_2 n]$-th orders. Specifically, we conducted experiments as follows. We trained AlexNet on the Tiny-ImageNet dataset to encourage interactions of specific orders without penalizing any interactions by setting $\lambda_1 = 1, \lambda_2 = 0$. Besides, we set $[r_1 = 0.2, r_2 = 0.5], [r_1 = 0.3, r_2 = 0.7]$, and $[r_1 = 0.6, r_2 = 0.9]$ in the $L^+(r_1, r_2)$ loss to learn three AlexNet models, respectively. We also trained AlexNet models to penalize interactions of specific orders by setting $\lambda_1 = 0, \lambda_2 = 1$. Two DNNs were trained by setting $[r_1 = 0, r_2 = 0.2]$ and $[r_1 = 0, r_2 = 0.5]$ in the $L^-(r_1, r_2)$ loss, respectively. Figure 4(b) shows the interaction strength $J^{(m)}$ of these DNNs. When we encouraged the DNN to encode interactions of the $[r_1 n, r_2 n]$-th orders, the interaction strength $J^{(m)}$ of the $[r_1 n, r_2 n]$-th orders significantly increased, compared to the normally trained DNNs. Figure 4(b) also shows that the loss $L^-(r_1, r_2)$ could successfully remove interactions of the $[r_1 n, r_2 n]$-th orders.

### 3.4 INVESTIGATION OF THE REPRESENTATION CAPACITIES

In the previous subsection, we introduced two losses, which force the DNN to encode interactions of different orders. In this subsection, we investigate the representation capacities of such DNNs. Thus, we conducted experiments to train four types of DNNs. The first type of DNN was normally trained. The other three types of DNNs were trained to mainly encode low-order, middle-order, and high-order interactions, respectively. Specifically, the second DNN was trained to penalize interactions of the $[0.7n, n]$-th orders by minimizing the $L^-(r_1, r_2)$ loss with $\lambda_1 = 0, \lambda_2 = 1, r_1 = 0.7, r_2 = 1.0$ [3]. The third DNN was learned to boost interactions of the $[0.3n, 0.7n]$-th orders by minimizing the $L^+(r_1, r_2)$ loss with $\lambda_1 = 1, \lambda_2 = 0, r_1 = 0.3, r_2 = 0.7$. The fourth DNN was trained to penalize interactions of the $[0, 0.5n]$-th orders by minimizing the $L^-(r_1, r_2)$ loss with $\lambda_1 = 0, \lambda_2 = 1, r_1 = 0, r_2 = 0.5$. The second DNN, the third DNN, and the fourth DNN were termed *the low-order DNN*, *the middle-order DNN* and *the high-order DNN*, respectively. In experiments, we trained three versions of each DNN based on the above four settings. We applied architectures of AlexNet and VGG-16/19 on the CIFAR-10 and the Tiny-ImageNet dataset.

Figure 4(c) and Figure 8 (in the appendix) show that the trained DNNs successfully learned interactions as expected. In other words, interactions of the $[0.7n, n]$-th orders were penalized in the low-order DNN. Interactions of the $[0.3n, 0.7n]$-th orders were boosted in the middle-order DNN. Interactions of the $[0, 0.5n]$-th orders were penalized in the high-order DNN.

**Classification accuracy.** Firstly, Table 1 shows classification performance of the above four types of DNNs. In general, the four types of DNNs achieved similar accuracies. The similar performance indicated that it was not necessary for a DNN to encode low-order interactions and high-order interactions to make inferences. Middle-order interactions could also provide discriminative information.

**Bag-of-words representations vs. structural representations.** Theoretically, high-order interactions usually represent the global structure of objects, which requires the complex collaborations of massive input variables. In comparison, low-order interactions learn local patterns from local and simple collaborations of a few input variables.

Therefore, we conducted two experiments to examine whether the high-order DNN encoded more structural information than the normally trained DNN. Specifically, as Figure 5 shows, in the first ex-

---

[3]The parameter $\lambda_2$ was set as 0.1 for VGG-16/19 networks trained on the Tiny-ImageNet dataset.

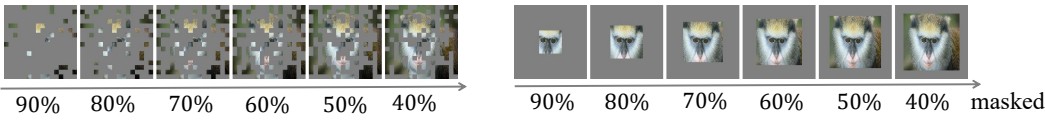

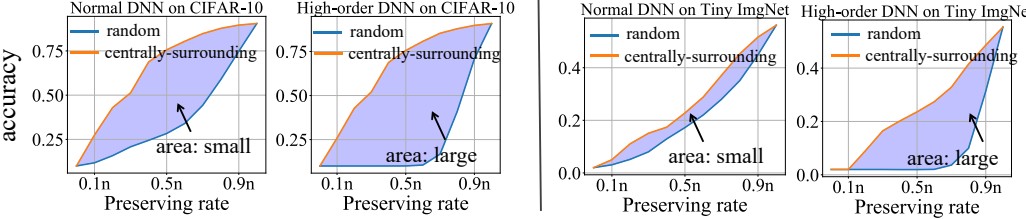

Figure 5: Tested images by random masking (left) and centrally-surrounding masking (right).

Figure 6: Classification accuracies using VGG-16 on images with different numbers of patches being masked. The Appendix C provides more results.

periment, we tested the DNN on images where $m$ patches in each image were randomly masked. In the second experiment, we tested the DNN on images where $m$ patches in each image on the image boundary were masked, and the patches in the center were preserved. In this way, we consider the structural information in tested images was destroyed in the first experiment, but such information was maintained in the second experiment. Here, in each sub-figure in Figures 6 and 9, we computed the area between the accuracy curve on samples generated by the random masking method and the accuracy curve on samples generated by the centrally-surrounding masking method. The area indicated the sensitivity to the structural destruction. We found that such area of high-order DNNs was much larger than the area of normally trained DNNs. This phenomenon indicated that normally trained DNN usually encoded local patterns, just like the bag-of-words representations, which were robust to the structural destruction. However, high-order DNN encoded more structural information.

**Adversarial robustness.** Ren et al. (2021b) have demonstrated that adversarial attacks mainly affected high-order interactions. Therefore, we conducted experiments to train DNNs mainly encoding high-order interactions based on the proposed losses, in order to verify whether such DNNs were more vulnerable to adversarial attacks. To train DNNs mainly encoding high-order interactions, we set $\lambda_1 = 1, \lambda_2 = 1$. Specifically, the DNN was trained to encourage interactions of the $[0.6n, n]$-th orders by setting $r_1 = 0.6, r_2 = 1$ for $L^+(r_1, r_2)$ and simultaneously penalize interactions of the $[0, 0.5n]$-th orders by setting $r_1 = 0, r_2 = 0.5$ for $L^-(r_1, r_2)$. We used the aforementioned MLP-5 and MLP-8 networks in Section 3.1. Each MLP was trained on the census and commercial datasets, respectively. Figure 7 in the appendix shows distributions of interaction strength of these DNNs. Then, we compared the adversarial robustness between normally trained DNNs and the DNNs whose high-order interactions were boosted. We adopted the untargeted PGD attack (Madry et al., 2018) based on $L_\infty$ norm. We set the attack strength $\epsilon = 0.6$ with 100 steps for the census dataset, and set $\epsilon = 0.2$ with 50 steps for the commercial dataset. The step size was uniformly set to 0.01 for all attacks. Please see more details in Appendix C. Table 1 shows that DNNs with boosted high-order interactions exhibited significantly lower adversarial accuracies than normally trained DNN, especially on the census dataset. These results verified that high-order interactions were vulnerable to adversarial attacks.

## 4 CONCLUSION

In this paper, we have discovered and theoretically proved the representation bottleneck of DNNs, from a new perspective of the complexity of interactions encoded in DNNs. We adopted the multi-order interaction, and used the order to represent the complexity of interactions. We discovered a common phenomenon that a DNN usually encoded very simple interactions and very complex interactions, but rarely learned interactions of intermediate complexity. We have theoretically proved the underlying reason for the representation bottleneck. Furthermore, we proposed two losses to learn DNNs which encoded interactions of specific complexities. Experimental results have shown that it is not necessary for a DNN to encode or avoid encoding interactions of specific orders, in terms of classification performance. However, high-order interactions usually encode more structural information than low-order interactions, and are usually vulnerable to adversarial attacks.

## 5 REPRODUCIBILITY STATEMENT

This research discovered and theoretically explained the representation bottleneck phenomenon, based on the multi-order interaction. Appendix A shows the trustworthiness of the multi-order interaction, by introducing its five desirable properties and its connections with existing typical metrics in game theory. Appendix B and Section 3.2 provide proofs for all theoretical results in the paper. Section 3 and Appendix C have discussed all experimental details, including the computation of interaction strength and how to train DNNs by the proposed two losses, which ensure the reproducibility. Furthermore, the code has been released at https://github.com/Nebularaid2000/bottleneck.

**Acknowledgments.** This work is partially supported by National Science and Technology Innovation 2030 Major Project of the Ministry of Science and Technology of China under Grant (2021ZD0111602), the National Nature Science Foundation of China (No. 61906120, U19B2043), Shanghai Natural Science Fundation (21JC1403800,21ZR1434600), Shanghai Municipal Science and Technology Major Project (2021SHZDZX0102).

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

## A   THE MULTI-ORDER INTERACTION

Zhang et al. (2020) proposed the multi-order interaction between input variables $i, j$ as follows:

$$I^{(m)}(i,j) = \mathbb{E}_{|S| \subseteq N \setminus \{i,j\}, |S|=m} \Delta v(i,j,S)$$

where $\Delta v(i,j,S) = v(S \cup \{i,j\}) - v(S \cup \{i\}) - v(S \cup \{j\}) + v(S)$. $I^{(m)}(i,j)$ denotes the interaction between variables $i, j \in N$ of the $m$-th order, which measures the average interaction utility between variables $i, j$ under contexts of $m$ variables. It has been proven that $I^{(m)}(i,j)$ satisfies the following five desirable properties.

•*Linear property.* If two independent games $u$ and $v$ are combined, *i.e.*, for $\forall S \subseteq N$, $w(S) = u(S) + v(S)$, then the multi-order interaction of the combined game equals to the sum of multi-order interactions derived from $u$ and $v$. *I.e.*, $I_w^{(m)}(i,j) = I_u^{(m)}(i,j) + I_v^{(m)}(i,j)$.

•*Nullity property.* A dummy variable $i \in N$ satisfies $\forall S \subseteq N \setminus \{i\}, v(S \cup \{i\}) = v(S) + v(\{i\})$. Then, the variable $i$ has no interactions with other variables, *i.e.*, $\forall m, \forall j \in N \setminus \{i\}, I^{(m)}(i,j) = 0$.

• *Commutativity property.* $\forall i, j \in N, I^{(m)}(i,j) = I^{(m)}(j,i)$.

• *Symmetry property.* Assume two variables $i, j$ are equivalent in the sense that $i, j$ have same cooperations with other variables, $\forall S \subseteq N \setminus \{i,j\}, v(S \cup \{i\}) = v(S \cup \{j\})$. Then, for any variable $k \in N, I^{(m)}(i,k) = I^{(m)}(j,k)$.

• *Efficiency property.* The network output of a DNN can be decomposed into the sum of interactions of different orders between different pairs of variables.

$$v(N) - v(\emptyset) = \sum_{i \in N} \mu_i + \sum_{i,j \in N, i \neq j} \sum_{m=0}^{n-2} w^{(m)} I^{(m)}(i,j).$$

where $\mu_i = v(\{i\}) - v(\emptyset)$ represents the independent effect of variable $i$, and $w^{(m)} = \frac{n-1-m}{n(n-1)}$.

**Connection with the Shapley value and the Shapley interaction index.**

*Shapley value.* Shapley (1951) proposed the Shapley value to measure the numerical importance of each player to the total reward in a cooperative game. The Shapley value has been widely used to explain the decision of DNNs in recent years (Lundberg & Lee, 2017; Ancona et al., 2019). Specifically, we can consider a DNN with a set of input variables $N = \{1, \ldots, n\}$ as a game. Each input variable (*e.g.*, an image pixel or a word) is regarded as a player, and the network output $v(N)$ of all input variables can be considered as the total reward of the game. The Shapley value aims to fairly distribute the network output to each individual variable as follows:

$$\phi(i) = \sum_{S \subseteq N \setminus \{i\}} \frac{|S|!(n-|S|-1)!}{n!} [v(S \cup \{i\}) - v(S)]$$

where $v(S)$ denotes the network output when we keep variables in $S$ unchanged while mask variables in $N \setminus S$ by the baseline value. The baseline value usually follows the setting in Ancona et al. (2019), which is set as the average value of the variable over different samples. In this way, $v(S \cup \{i\}) - v(S)$ represents the marginal contribution of $i$ when the variable $i$ is present *w.r.t.* the case when the variable $i$ is absent, given the context $S \subseteq N \setminus \{i\}$. Then, the Shapley value of $i$ measures the average marginal contribution of $i$ over different contexts $S$. It has been proven that the Shapely value is the unique method to fairly allocate overall reward to each player that satisfies *linearity*, *nullity*, *symmetry*, and *efficiency* properties.

*Shapley interaction index.* Grabisch & Roubens (1999) proposed the Shapley interaction index $I(S)$ to measure the interaction utility between input variables in the subset $S \subseteq N$. In particular, the Shapley interaction index between two variables $i, j \in N$, $Index(i,j)$, measures the change of the numerical importance (*i.e.*, Shapley value) of $i$ by the presence or absence of $j$.

$$Index(i,j) = \tilde{\phi}(i)_{j \text{ always present}} - \tilde{\phi}(i)_{j \text{ always absent}}, \tag{10}$$

where $\tilde{\phi}(i)_{j \text{ always present}}$ denotes the Shapley value of the variable $i$ computed under the specific condition that the variable $j$ is always present. $\tilde{\phi}(i)_{j \text{ always absent}}$ is computed under the specific condition that $j$ is always absent.

**Connections with the Shapley interaction index and the Shapley value.** We found that the multi-order interaction has strong connections with the Shapley interaction index and the Shapley value.

Specifically, it has been proven that the interaction index $Index(i, j)$ between variables $i, j$, which is closely related to the Shapley interaction index, can be decomposed into multi-order interactions as follows.

$$Index(i, j) = \frac{1}{n-1} \sum_{m=0}^{n-2} I^{(m)}(i, j).$$

Besides, the Shapley value of the variable $i$ also can be decomposed into the multi-order interactions.

$$\phi(i) = \frac{1}{n} \sum_{m=1}^{n-1} \mathbb{E}_{j \in N \setminus \{i\}} \left[ \sum_{k=0}^{m-1} I^{(k)}(i, j) \right] + v(i) - v(\emptyset)$$

# B    PROOF OF THEOREMS

## B.1    PROOF OF THEOREM 1

**Motivation of Theorem 1.** Let $W$ denote the network parameters of the DNN. Let $L$ and $\eta$ denote the loss function and learning rate of training, respectively. Here, we consider the change of network parameters $\Delta W = -\eta \frac{\partial L}{\partial v(N)} \frac{\partial v(N)}{\partial W}$, whose norm indicates the learning strength of the DNN. According to the efficiency property of $I^{(m)}(i, j)$ in Eq. (2), the network output $v(N)$ of a DNN can be decomposed into the sum of multi-order interactions $I^{(m)}(i, j)$ of different orders between different pairs of variables. Therefore, the change $\Delta W$ of parameters can be also decomposed into the sum of the gradient of multi-order interactions *w.r.t.* the parameters, *i.e.*, $\frac{\partial I^{(m)}(i,j)}{\partial W}$. Specifically,

$$\Delta W = \Delta W_U + \sum_{m=0}^{n-2} \sum_{i,j \in N, i \neq j} \Delta W^{(m)}(i, j)$$

where $U = v(\emptyset) + \sum_{i \in N} \mu_i$. And,

$$\Delta W_U \stackrel{\text{def}}{=} -\eta \frac{\partial L}{\partial v(N)} \frac{\partial v(N)}{\partial U} \frac{\partial U}{\partial W}, \quad \Delta W^{(m)}(i, j) \stackrel{\text{def}}{=} R^{(m)} \frac{\partial I^{(m)}(i, j)}{\partial W}.$$

where $R^{(m)} = -\eta \frac{\partial L}{\partial v(N)} \frac{\partial v(N)}{\partial I^{(m)}(i,j)}$. Based on the analysis in Section 3.2, the term $\Delta W^{(m)}(i, j) = R^{(m)} \frac{\partial I^{(m)}(i,j)}{\partial W}$ represents the strength of learning the $m$-order interactions.

Thus, we aim to study the strength $\Delta W^{(m)}(i, j)$ of learning the $m$-order interaction in Theorem 1.

**Proof skeleton in Theorem 1.** To theoretically explain the representation bottleneck, we prove that the strength of learning middle-order interactions (*i.e.*, $\Delta W^{(m)}(i, j), m \approx 0.5n$) is much smaller than the strength of learning low-order and high-order interactions. It is because that the learning strength of middle-order interactions is an average of the learning gradients of $\Delta v(i, j, S)$s (*i.e.*, $\frac{\partial \Delta v(i,j,S)}{\partial W}$) **over massive contexts** $S$, which results in cancellation of these gradients. In contrast, the learning strength of low-order interactions (high-order interactions) is the average of $\frac{\partial \Delta v(i,j,S)}{\partial W}$s **over a few contexts** $S$, which mitigates the cancellation phenomenon. Thus, DNNs are more likely to encode low-order and high-order interactions, but usually fail to encode middle-order interactions.

**Proof of Theorem 1.** The $m$-order interaction $I^{(m)}(i, j)$ between variables $i, j$ is defined as,

$$\begin{aligned}
I^{(m)}(i, j) &= \mathbb{E}_{S \subseteq N \setminus \{i,j\}, |S|=m} \Delta v(i, j, S) \\
&= \frac{1}{\binom{n-2}{m}} \sum_{\substack{S \subseteq N \setminus \{i,j\} \\ |S|=m}} \Delta v(i, j, S).
\end{aligned} \tag{11}$$

We use $W = [W_1, W_2, \ldots, W_K]^\top \in \mathbb{R}^K$ to denote the network parameters. Then, based on the Eq. (11),

$$\begin{aligned}
\frac{\partial I^{(m)}(i, j)}{\partial W} &= \sum_{\substack{S \subseteq N \setminus \{i,j\} \\ |S|=m}} \frac{\partial I^{(m)}(i, j)}{\partial \Delta v(i, j, S)} \frac{\partial \Delta v(i, j, S)}{\partial W} \\
&= \frac{1}{\binom{n-2}{m}} \sum_{\substack{S \subseteq N \setminus \{i,j\} \\ |S|=m}} \frac{\partial \Delta v(i, j, S)}{\partial W}.
\end{aligned}$$

Assume $\mathbb{E}_{i,j,S}[\frac{\partial \Delta v(i,j,S)}{\partial W}] = \mathbf{0}$. Without loss of generality, let $\sigma^2$ denote the variance of each dimension of $\frac{\partial \Delta v(i,j,S)}{\partial W}$. Since the gradients $\frac{\partial \Delta v(i,j,S)}{\partial W}$ on different contexts are independent with each other, then we have

$$\mathbb{E}_{i,j}[\frac{\partial I^{(m)}(i,j)}{\partial W}] = \mathbf{0},$$

$$\mathrm{Var}_{i,j}[\frac{\partial I^{(m)}(i,j)}{\partial w_k}] = \sigma^2/\binom{n-2}{m}, \ \forall k = 1,\ldots,K.$$

where $K$ is the dimension of network parameters $W$. Furthermore, because $\Delta W^{(m)}(i,j) = -\eta \frac{\partial L}{\partial v(N)} w^{(m)} \frac{\partial I^{(m)}(i,j)}{\partial W}$, then

$$\mathbb{E}_{i,j}[\Delta W^{(m)}(i,j)] = \mathbf{0},$$

$$\mathrm{Var}_{i,j}[\Delta W_k^{(m)}(i,j)] = (\eta \frac{\partial L}{\partial v(N)} w^{(m)})^2 \sigma^2/\binom{n-2}{m}, \ \forall k = 1,\ldots,K.$$

where $w^{(m)} = \frac{n-m-1}{n(n-1)}$, and $\Delta W_k^{(m)}(i,j) = -\eta \frac{\partial L}{\partial v(N)} w^{(m)} \frac{\partial I^{(m)}(i,j)}{\partial w_k}$ represents the $k$-th dimension of $\Delta W^{(m)}(i,j)$. Moreover, we can obtain,

$$
\begin{aligned}
\mathbb{E}_{i,j}[\|\Delta W^{(m)}(i,j)\|_2^2] &= \mathbb{E}_{i,j}[\sum_{k=1}^{K} \Delta W_k^{(m)}(i,j)^2] = \sum_{k=1}^{K} \mathbb{E}_{i,j}[\Delta W_k^{(m)}(i,j)^2] \\
&= \sum_{k=1}^{K}[(\mathbb{E}_{i,j}[\Delta W_k^{(m)}(i,j)])^2 + \mathrm{Var}_{i,j}[\Delta W_k^{(m)}(i,j)]] \\
&= \sum_{k=1}^{K} \mathrm{Var}_{i,j}[\Delta W_k^{(m)}(i,j)] \\
&= K(\eta \frac{\partial L}{\partial v(N)} \frac{n-m-1}{n(n-1)})^2 \sigma^2/\binom{n-2}{m}.
\end{aligned}
\tag{12}
$$

Therefore, the conclusion of Theorem 1 holds.

## B.2 PROOF OF THEOREM 2

Let $S_1, S_2$ denote the two variable subsets randomly sampled from the universal set $N$ including all input variables, where $|S_1| = r_1 n, |S_2| = r_2 n$, and $0 \le r_1 < r_2 \le 1$.

When we consider each $S_1$ as the universal set, according to the efficiency property of the multi-order interaction, we can obtain,

$$
\begin{aligned}
\mathbb{E}_{S_1}[v(S_1)] &= v(\emptyset) + \mathbb{E}_{S_1}[\sum_{i \in S_1} \mu_i] + \mathbb{E}_{S_1}[\sum_{i,j \in S_1, i \neq j}[\sum_{m=0}^{r_1 n - 2} \frac{r_1 n - 1 - m}{r_1 n(r_1 n - 1)} I_{S_1}^{(m)}(i,j)]] \\
&= v(\emptyset) + r_1 n \mathbb{E}_i(\mu_i) + \mathbb{E}_{S_1}[\sum_{m=0}^{r_1 n - 2} \frac{r_1 n - 1 - m}{r_1 n(r_1 n - 1)} \sum_{i,j \in S_1, i \neq j} I_{S_1}^{(m)}(i,j)] \\
&= v(\emptyset) + r_1 n \mathbb{E}_i(\mu_i) + \sum_{m=0}^{r_1 n - 2} (r_1 n - 1 - m) \mathbb{E}_{S_1}[\mathbb{E}_{i,j}[(I_{S_1}^{(m)}(i,j))]]
\end{aligned}
$$

where $I_{S_1}^{(m)}(i,j) \stackrel{\text{def}}{=} \mathbb{E}_{S' \subseteq S_1 \setminus \{i,j\}, |S'|=m}(\Delta v(i,j,S'))$.

Similarly, when we consider each $S_2$ as the universal set, we can obtain,

$$
\begin{aligned}
\mathbb{E}_{S_2}[v(S_2)] &= v(\emptyset) + \mathbb{E}_{S_2}[\sum_{i \in S_2} \mu_i] + \mathbb{E}_{S_2}[\sum_{i,j \in S_2, i \neq j}[\sum_{m=0}^{r_2 n - 2} \frac{r_2 n - 1 - m}{r_2 n(r_2 n - 1)} I_{S_2}^{(m)}(i,j)]] \\
&= v(\emptyset) + r_2 n \mathbb{E}_i(\mu_i) + \sum_{m=0}^{r_2 n - 2} (r_2 n - 1 - m) \mathbb{E}_{S_2}[\mathbb{E}_{i,j}[(I_{S_2}^{(m)}(i,j))]]
\end{aligned}
$$

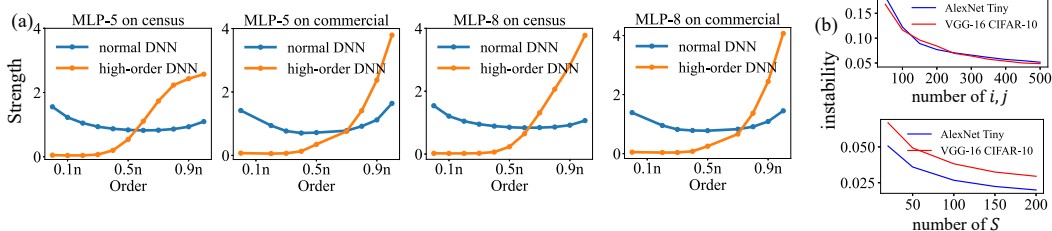

Figure 7: (a) Distributions of the interaction strength $J^{(m)}$ of normal DNNs and high-order DNNs, where high-order DNNs were trained by encouraging high-order interactions and penalizing low-order interactions simultaneously. (b) The instability of $J^{(m)}(x)$ *w.r.t.* the sampling number of pairs of variables $(i, j)$ and the sampling number of contexts $S$.

where $I_{S_2}^{(m)}(i, j) \overset{\text{def}}{=} \mathbb{E}_{S' \subseteq S_2 \setminus \{i,j\}, |S'|=m}(\Delta v(i, j, S'))$. Note that the contexts when computing $I_{S_1}^{(m)}(i, j)$ and $I_{S_2}^{(m)}(i, j)$ are different. It is easy to obtain that

$$\mathbb{E}_{S_1}[\mathbb{E}_{i,j}(I_{S_1}^{(m)}(i, j))] = \mathbb{E}_{S_2}[\mathbb{E}_{i,j}(I_{S_2}^{(m)}(i, j))] = \mathbb{E}_{i,j}(I^{(m)}(i, j)).$$

When $m < r_1 n < r_2 n$. Therefore, $\Delta u(r_1, r_2)$ can be rewritten as follows:

$$
\begin{aligned}
\Delta u(r_1, r_2) &= \mathbb{E}_{(S_1, S_2): \emptyset \subseteq S_1 \subsetneq S_2 \subseteq N}[v(S_2) - r_2/r_1 \cdot v(S_1)] \\
&= \mathbb{E}_{S_2}[\mathbb{E}_{S_1}[v(S_2) - r_2/r_1 \cdot v(S_1)]] \\
&= \mathbb{E}_{S_2}[v(S_2)] - r_2/r_1 \mathbb{E}_{S_1}[v(S_1)] \\
&= (1 - r_2/r_1)v(\emptyset) + \sum_{m=0}^{n-2} \sum_{i,j \in N, i \neq j} \tilde{w}^{(m)} I^{(m)}(i, j)
\end{aligned}
$$

$$
\text{where} \quad \tilde{w}^{(m)} = \begin{cases} (r_2/r_1 - 1)(m+1)/[n(n-1)], & m \leq r_1 n - 2 \\ (r_2 n - m - 1)/[n(n-1)], & r_1 n - 2 < m \leq r_2 n - 2 \\ 0, & r_2 n - 2 < m \leq n - 2 \end{cases}
$$

Then, the conclusion holds.

## C  EXPERIMENTAL DETAILS AND MORE RESULTS

### C.1  IMPLEMENTATION DETAILS

The experiments were conducted on the CIFAR-10, Tiny-ImageNet, ImageNet, and two tabular datasets. Due to the computational cost, we selected 50 classes from 200 classes at equal intervals (*i.e.*, the 4th, 8th, ..., 196th, 200th classes) when we trained DNNs on the Tiny-ImageNet dataset.

**The sampling strategy in the computation of $J^{(m)}$.** The interaction strength $J^{(m)}$ is defined as,

$$J^{(m)} = \frac{\mathbb{E}_{x \in \Omega}[\mathbb{E}_{i,j}[|I^{(m)}(i, j|x)|]]}{\mathbb{E}_{m'}[\mathbb{E}_{x \in \Omega}[\mathbb{E}_{i,j}[|I^{(m')}(i, j|x)|]]]}, \text{ where } I^{(m)}(i, j|x) = \mathbb{E}_{\substack{S \subseteq N \setminus \{i,j\} \\ |S|=m}}[\Delta v(i, j, S)].$$

To precisely compute $J^{(m)}$, we need to average all possible contexts $S \subseteq N$, all pairs of variables $i, j \in N$, and all samples $x \in \Omega$, which is usually computationally infeasible. Therefore, we adopted the sampling strategy used in (Zhang et al., 2020) to approximately compute $J^{(m)}$.

The sampling strategy was conducted as follows. With respect to the sampling number of input samples, we sampled 50 correctly classified samples on image datasets. On the ImageNet and the Tiny-ImageNet dataset, these images were sampled from different 50 classes. On the CIFAR-10 dataset, we sampled 5 images from each class. Then, for each image, we sampled 200 pairs of patches $i, j \in N$. Since DNNs usually encode stronger interactions between neighbor patches, we restricted that patch $i$ should be located at the neighborhood patch $j$ with a radius of two patches. Next, for each pair of patches $i, j$. and each order $m$, we randomly sampled 100 contexts $S$ from all possible contexts, where $|S| = m$. Besides, we computed 13 different orders for $J^{(m)}$, where $m = 0, 0.05n, 0.1n, 0.2n, ..., 0.8n, 0.9n, 0.95n, 1.0n$. On each tabular dataset, since there are only

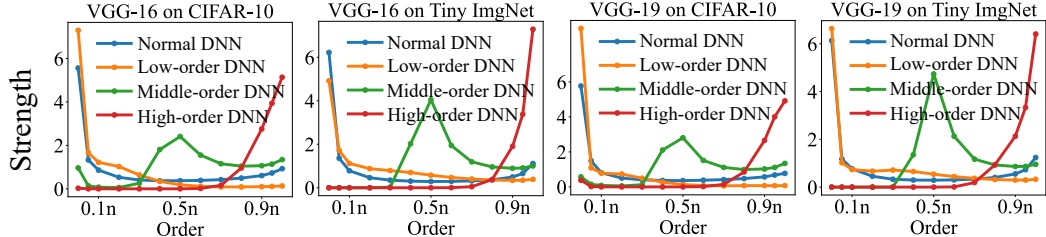

Figure 8: Distributions of the interaction strength $J^{(m)}$ of four types of DNNs. The low-order, middle-order, and high-order DNNs were trained by following the parameter setting mentioned in Section 3.4, while $\lambda_2$ was set as 0.1 (instead of 1) for low-order DNNs on the Tiny-ImageNet dataset.

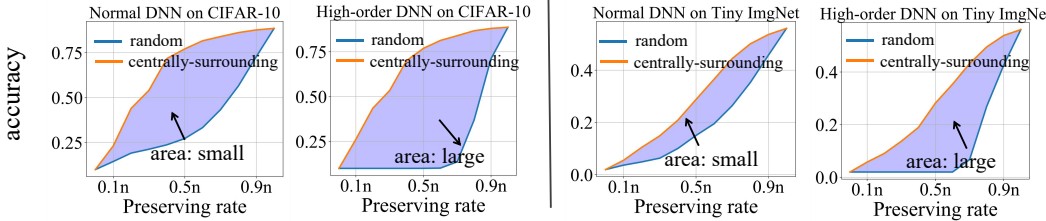

Figure 9: Classification accuracies using AlexNet architecture on images with different numbers of patches being masked.

a few input variables ($n = 12$ on the census dataset and $n = 10$ on the commercial dataset), we sampled 100 instances and all pairs of variables for computation. For each pair of patches and each order $m$, we randomly sampled 100 contexts $S$ from all possible contexts.

To validate the reliability of the approximated $J^{(m)}$ via the above sampling strategy, we evaluated the (in)stability of $J^{(m)}$ during the sampling process. Specifically, we designed an instability metric when $J^{(m)}$ was repeatedly computed for $q$ times. Firstly, for each input image $x$, we defined $J^m(x) \stackrel{\text{def}}{=} \frac{\mathbb{E}_{i,j}[|I^{(m)}(i,j|x)|]}{\mathbb{E}_{m'}\mathbb{E}_{i,j}[|I^{(m)}(i,j|x)|]}$. The instability *w.r.t.* the $J^{(m)}(x)$ was computed as $\frac{\mathbb{E}_{u,v;u\neq v}|J_u^{(m)}(x)-J_v^{(m)}(x)|}{\mathbb{E}_w|J_w^{(m)}(x)|}$, where $J_u^{(m)}(x)$, $J_v^{(m)}(x)$ denote the estimated $J^{(m)}(x)$ at the $u$-th sampling time and at the $v$-th sampling time, respectively. Then, the instability of $J^{(m)}$ was computed as follows.

$$instability = \mathbb{E}_{x\in\Omega}\mathbb{E}_m\left[\frac{\mathbb{E}_{u,v;u\neq v}|J_u^{(m)}(x)-J_v^{(m)}(x)|}{\mathbb{E}_w|J_w^{(m)}(x)|}\right].$$

The instability is an average over all sampled images and all sampled orders.

Based on the above definition, we used the trained AlexNet on the Tiny-ImageNet dataset and VGG-16 on the CIFAR-10 dataset to compute the above instability. We conducted two experiments to evaluate the instability of $J^{(m)}$ *w.r.t* the sampling of the contexts $S$ and the sampling of pairs $(i,j)$, respectively. In the first experiment, we fixed 100 pairs of $(i,j)$ and evaluated the instability of $J^{(m)}$ *w.r.t* the sampling of the contexts $S$. Figure 7(b) shows that on the two datasets, when the sampling number of $S$ increased, the instability decreased. Furthermore, when the sampling number of $S$ was greater than 100 (*i.e.,* our setting), the instability value was less than 0.05, which indicated a stable approximation. In the second experiment, we evaluated the instability *w.r.t* the sampling processes of $(i,j)$, where the sampling number of S was set to 100 as above-mentioned. As Figure 7(b) shows, when the sampling number of $(i,j)$ pairs increased, the instability decreased. When the sampling number of $(i,j)$ pairs was greater than 200 (*i.e.,* our setting), the instability was less than 0.1, which indicated a stable approximation of $J^{(m)}$. These results demonstrated that the adopted sampling strategy could well approximate the interaction strength $J^{(m)}$.

**Implementation details of adversarial attacks.** Here, we introduce how to measure the adversarial robustness in Section 3.4. We adopted the untargeted PGD attack (Madry et al., 2018) with the $L_\infty$ constraint $\|\Delta x\|_\infty \leq \epsilon$ to generate adversarial examples. For the census dataset, we set $\epsilon = 0.6$ and

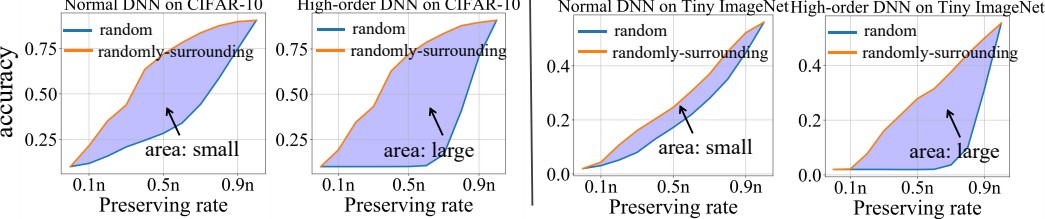

Figure 10: Images generated by random masking (left) and randomly-surrounding masking (right).

Figure 11: Classification accuracies using VGG-16 architecture on images with different numbers of patches being masked (by random masking and randomly-surrounding masking).

the attack was conducted with 100 steps. For the commercial dataset, we set $\epsilon = 0.2$ and the attack was conducted with 50 steps. The step size was set to 0.01 for all attacks.

## C.2 MORE EXPERIMENTAL RESULTS.

In this subsection, we provide more experimental results besides the results in the main paper.

### C.2.1 MORE EXPERIMENTS ON THE EFFECTIVENESS OF LOSSES

Except for the AlexNet architecture, we also used the proposed losses to train four types of DNNs of VGG-16 and VGG-19 architectures, *i.e.*, normally trained DNNs, low-order DNNs, middle-order DNNs, and high-order DNNs, on the CIFAR-10 dataset and the Tiny-ImageNet dataset. The parameters were set as the same in Section 3.4. Note that when $r_1 = 0$, we set $\Delta u(r_1, r_2) = \mathbb{E}_{S_2}[v(S_2)] - v(\emptyset)$. Experimental results in Figure 8 demonstrated that the four types of DNNs usually could successfully learn interactions as expected, which further validated the effectiveness of the proposed losses.

### C.2.2 MORE EXPERIMENTS ON STRUCTURAL REPRESENTATION

**Verify structural representations on more DNNs**. Figure 9 shows classification accuracies using AlexNet network on images with different numbers of patches being masked. These results further validated that high-order DNNs were more sensitive to the destruction of structural information.

**More masking methods to verify structural representations.** Besides, we designed a new masking method to further investigate a DNN's capacity of encoding structural information. Specifically, we masked $m$ patches in each image and only preserved a single large region of $n - m$ patches, as shown in Figure 10. The position of the preserved region was randomly determined, instead of being fixed in the center of the image previously. We called the new masking method as *randomly-surrounding masking method*. By doing so, the structural information within the large region was maintained.

Figures 11 and 12 show experimental results. In each sub-figure, the $x$-axis represents the ratio of preserved patches, and the $y$-axis represents the classification accuracy when only the masked images were fed as input. In addition, the blue curve denotes the accuracy curve on the samples generated by the random masking method, and the orange curve shows the accuracy on the samples generated by the randomly-surrounding masking method. The area between the blue curve and the orange curve measures the sensitivity of the DNN to the structural destruction.

Based on the accuracy on masked samples in Figures 11 and 12, we found that the aforementioned area of the high-order DNN was much larger than the area of the normally trained DNN. The experimental results verified that high-order DNNs encoded more structural information than normally trained DNNs. Such results were also consistent with results obtained in our previous experiments in Figures 6 and 9.

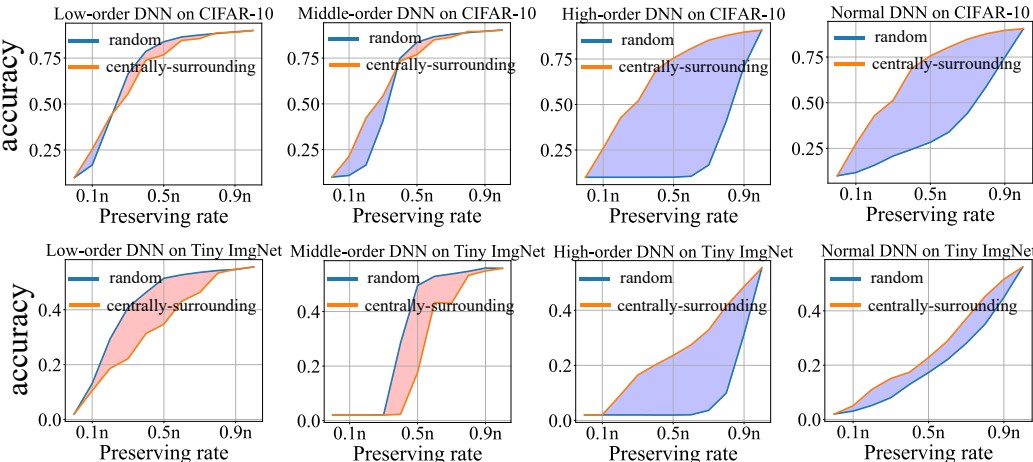

Figure 12: Classification accuracies using AlexNet architecture on images with different numbers of patches being masked (by random masking and randomly-surrounding masking).

Figure 13: Classification accuracies of low-order DNNs, middle-order DNNs, high-order DNNs, and normally trained DNNs (using VGG16) on images with different numbers of patches being masked. Here, we used the random masking method and the centrally-surrounding masking method. The blue color indicated that the random masking method had a lower accuracy, while the red color indicated that the random masking method had a higher accuracy. The phenomenon that the random masking method had a higher accuracy may be because the random masking method preserved more abundant local information in generated images, while the centrally-surrounding masking method will make the unmasked regions concentrate together. Considering that adjacent regions often contain similar feature information, the centrally-surrounding masking method will lead to redundancy of information and limit the diversity of information.

**Comparisons of structural representations between four types of DNNs.** We further compared the capacity of encoding structural representations among four types of DNNs (*i.e.*, normally trained DNNs, low-order DNNs, middle-order DNNs, and high-order DNNs). To this end, we followed the experimental setting in Section 3.4. We trained low-order DNNs by penalizing interactions of the $[0.7n, n]$-th orders based on $L^-(r_1, r_2)$ loss, and trained middle-order DNNs by boosting interactions of the $[0.3n, 0.7n]$-th orders based on $L^+(r_1, r_2)$ loss.

Figure 13 compares classification accuracies on images generated by the random masking method (which were considered as bag-of-words features without structures) and images generated by the centrally-surrounding masking method (with structural patterns). The large gap between the two classification accuracies indicated that the learned DNN encoded rich structural information for inference. *I.e.*, the larger area between the accuracy curve on images generated by the random masking method and the accuracy curve on images generated by the centrally-surrounding masking method means the richer structural information in the DNN. In this way, Figure 13 compares the capacity of encoding structural information between four types of VGG-16 networks. Similarly, Figure 14 compares the capacity of encoding structural information between four types of AlexNets.

To this end, we obtained three conclusions.

(i) Low-order DNNs trained on the CIFAR-10 dataset encoded little structural information, because such DNNs did not exhibit much difference in classification accuracy on the above two types of

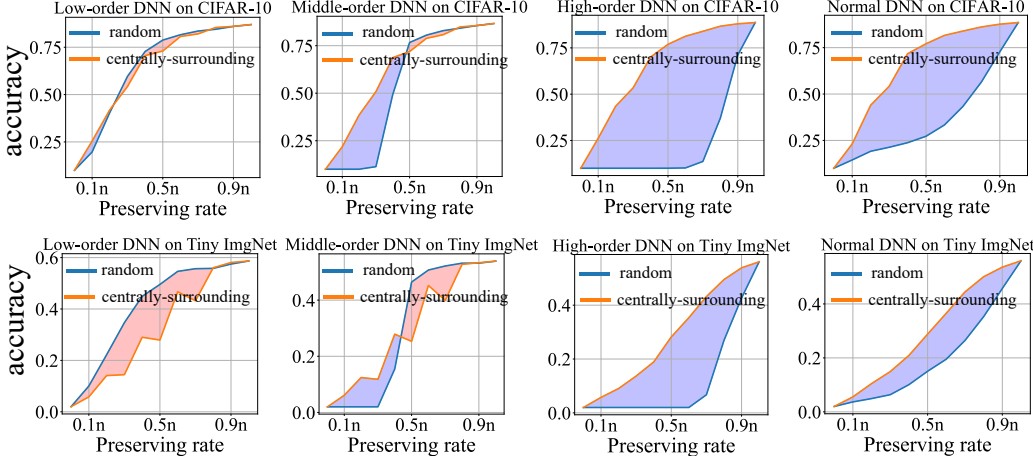

Figure 14: Classification accuracies of low-order DNNs, middle-order DNNs, high-order DNNs, and normally trained DNNs (using AlexNet) on images with different numbers of patches being masked. Here, we used the random masking method and the centrally-surrounding masking method. The blue color indicated that the random masking method had a lower accuracy, while the red color indicated that the random masking method had a higher accuracy. The phenomenon that the random masking method had a higher accuracy may be because the random masking method preserved more abundant local information in generated images, while the centrally-surrounding masking method will make the unmasked regions concentrate together. Considering that adjacent regions often contain similar feature information, the centrally-surrounding masking method will lead to redundancy of information and limit the diversity of information.

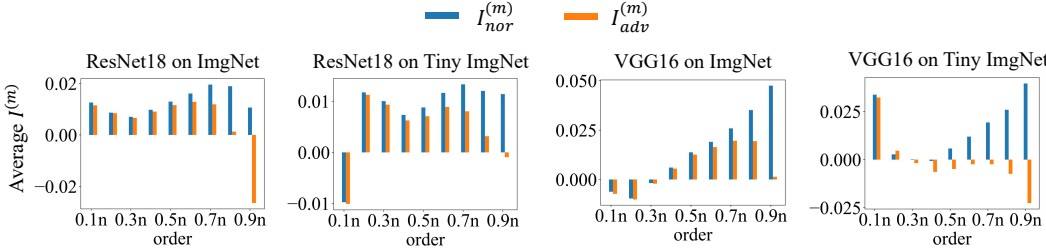

Figure 15: The multi-order interaction $I_{nor}^{(m)}$ on normal samples and $I_{adv}^{(m)}$ on adversarial samples for normally trained DNNs. Adversarial perturbations mainly affected high-order interactions.

samples. Furthermore, for low-order DNNs trained on the Tiny-ImageNet dataset, the accuracies on images generated by the random masking method were even higher than the accuracies on images generated by the centrally-surrounding masking method. This may be because the random masking method preserved more abundant local information in generated images, while the centrally-surrounding masking method will make the unmasked regions concentrate together. Considering that adjacent regions often contain similar feature information, the centrally-surrounding masking method will lead to redundancy of information and limit the diversity of information. This result further verified that low-order DNNs preferred bag-of-words representations, but failed to encode structural information.

(ii) High-order DNNs showed the highest accuracy difference between the above two types of samples, which indicated that high-order DNNs encoded most structural information.

(iii) The capacity of Middle-order DNNs to encode structural information was somewhere in between low-order DNNs and high-order DNNs. It indicated that middle-order DNNs encoded more structural information than low-order DNNs, but middle-order DNNs encoded less structural information than high-order DNNs.

Experimental results demonstrated that it was the high-order interaction that was mainly responsible for encoding structural information.

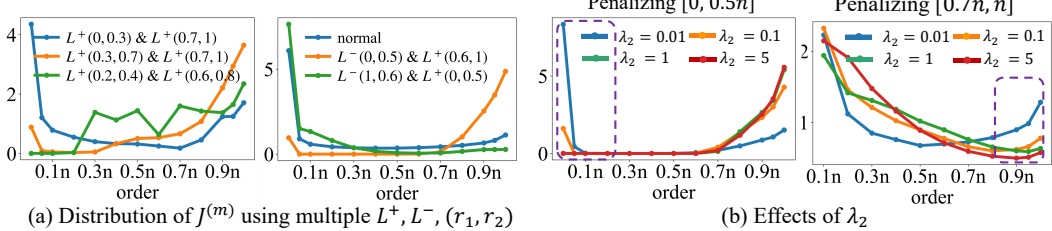

Figure 16: (a) Distributions of the interaction strength $J^{(m)}$ of normally trained DNN and DNNs trained by using multiple $L^+$, $L^-$, and $(r_1, r_2)$ pairs. (b) Distributions of the interaction strength $J^{(m)}$ of DNNs trained by using different $\lambda_2$s.

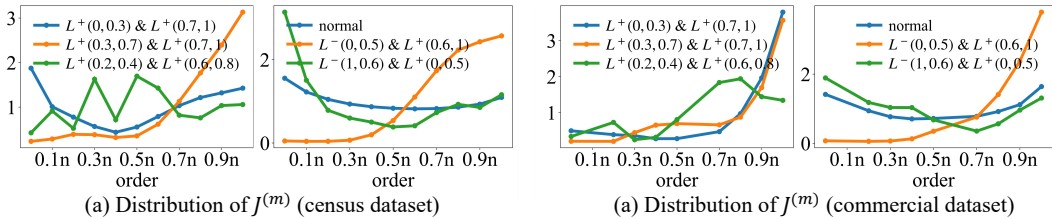

Figure 17: Distributions of the interaction strength $J^{(m)}$ of normally trained DNN and DNNs trained with $L^+$ and/or $L^-$ w.r.t. different pairs of $(r_1, r_2)$.

### C.2.3 MORE EXPERIMENTS ON ADVERSARIAL ROBUSTNESS

**Comparisons on adversarial robustness between four types of DNNs.** We further compared the adversarial robustness between four types of DNNs, including low-order DNNs, middle-order DNNs, high-order DNNs, and normally trained DNNs. To this end, we followed the experimental setting in Section 3.4. Here, the low-order DNNs were trained by encouraging interactions of the $[0, 0.5n]$-th orders based on $L^+(r_1, r_2)$ and simultaneously penalizing interactions of the $[0.6n, n]$-th orders based on $L^-(r_1, r_2)$. We set $\lambda_1 = 10, \lambda_2 = 10$. The middle-order DNNs were trained to encourage interactions of the $[0.3n, 0.7n]$-th orders based on $L^+(r_1, r_2)$. We set $\lambda_1 = 10, \lambda_2 = 0$. We used the MLP-5 and MLP-8 networks in Section 3.1. Each MLP was trained on the census dataset and commercial dataset, respectively. Adversarial attacks were conducted by following experimental settings in Section 3.4.

Table 2 reports the adversarial robustness of four types of DNNs. We obtained following results. (i) Low-order DNNs usually exhibited the highest adversarial robustness, which outperformed normally trained DNNs. (ii) High-order DNNs exhibited the lowest robustness, which was significantly lower than normally trained DNNs. (iii) The robustness of middle-order DNNs was somewhere in between low-order and high-order DNNs. The above results showed a significant connection between adversarial robustness and the orders of the encoded interactions.

**More explorations on Adversarial robustness.** We further explored the relationship between adversarial robustness and the order of interactions. We conducted experiments in Ren et al. (2021b) on images in the ImageNet dataset and the Tiny-ImageNet dataset, in order to explore interactions of which orders were vulnerable to adversarial attacks.

Specifically, let $x$ denote a normal sample, and let $x' = x + \epsilon$ denote an adversarial sample. Then, according to the efficiency property of $I^{(m)}(i, j)$, the significant output change caused by adversarial perturbations, $v(N|x) - v(N|x')$, can be represented as the sum of differences in multi-order interactions as follows.

$$v(N|x) - v(N|x') = \text{ignorable bias} + \sum_m w^{(m)} \sum_{i,j} \Delta I^{(m)}(i, j),$$

where $\Delta I^{(m)}(i, j) = I^{(m)}(i, j|x) - I^{(m)}(i, j|x')$ is the compositional adversarial effect on the interaction utility $I^{(m)}(i, j|x)$ caused by adversarial perturbations. In this way, the overall attacking utility (the output change) can be decomposed into adversarial effects on massive compositional interactions $I^{(m)}(i, j|x)$.

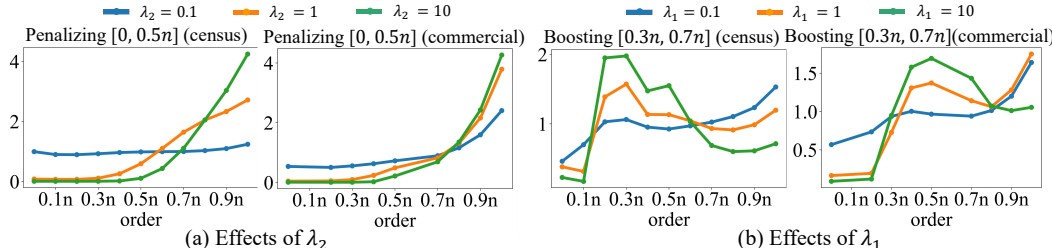

Figure 18: Distributions of the interaction strength $J^{(m)}$ of DNNs trained by using different pairs $(\lambda_1, \lambda_2)$ on the census dataset and the commercial dataset.

Table 2: Comparison of the adversarial accuracy between normally trained DNNs, low-order DNNs, middle-order DNNs, and high-order DNNs on the census dataset and the commercial dataset.

| Model | Normal training | penalize high-order & boost low-order | boost middle-order | Penalize low-order & boost high-order |
|---|---|---|---|---|
| MLP-5 on census | 38.22 | 42.40 | 15.93 | 7.31 |
| MLP-8 on census | 39.33 | 44.65 | 18.10 | 2.02 |
| MLP-5 on commercial | 27.01 | 29.76 | 23.70 | 22.00 |
| MLP-8 on commercial | 25.92 | 28.86 | 22.55 | 20.58 |

Therefore, Figure 15 uses the change of multi-order interactions $\Delta I^{(m)}(i, j)$ as specific reasons for adversarial vulnerability. This figure shows the difference between interactions of normal samples and interactions of adversarial samples. Adversarial perturbations mainly affected high-order interactions, and rarely affected low-order interactions. This phenomenon has been showed in some previous studies. The above experiments further verified the connection between adversarial robustness and the order of the encoded interactions.

In summary, experimental results showed a strong connection between adversarial robustness and the order of the encoded interactions. Nevertheless, the order of interactions does not fully determine the adversarial robustness. For example, previous studies have shown that interactions encoded in adversarially trained DNNs were also more robust to attacks.

### C.2.4 MORE EXPERIMENTS ON DIFFERENT HYPER-PARAMETERS

**Training DNNs with both $L^+$ and $L^-$ *w.r.t.* different pairs of $(r_1, r_2)$.** We investigated the performances of DNNs when we simultaneously used multiple $L^+$ and $L^-$ with different pairs of $(r_1, r_2)$ in the loss function. We used the following five sets of experimental settings, including (i) $L^+(0, 0.3)$ and $L^+(0.7, 1.0)$; (ii) $L^+(0.3, 0.7)$ and $L^+(0.7, 1.0)$; (iii) $L^+(0.2, 0.4)$ and $L^+(0.6, 0.8)$; (iv) $L^-(0, 0.5)$ and $L^+(0.6, 1)$; (v) $L^+(0, 0.5)$ and $L^-(0.6, 1)$ to learn DNNs. DNNs were trained on the CIFAR-10 dataset, the census dataset, and the commercial dataset, respectively.

Figure 16(a) shows that the trained DNNs successfully learned interactions as expected. For example, the DNN trained with $L^+(0, 0.3)$ & $L^+(0.7, 1.0)$ simultaneously boosted both interactions of the $[0, 0.3n]$-orders and interactions of the $[0.7n, n]$-orders. These results further verified the effectiveness of the proposed losses.

Furthermore, DNNs trained with $L^+$ and/or $L^-$ *w.r.t.* different pairs of $(r_1, r_2)$ further verified the conclusion obtained in Table 1 (left). *I.e.*, although the interactions of different orders performed differently in both adversarial robustness (Table 1 (right)) and the capacity of encoding structural information (Figures 6, 9, 13, and 14), interactions of different orders had similar classification accuracies (the difference was within $\pm 1\%$ accuracy). It showed that although interactions of different orders had their distinctive properties, such difference did not necessarily affect their classification accuracies on normal images.

**Effects of hyper-parameters.** We further investigated the effects of different hyper-parameters.

First, the effects of a single $(r_1, r_2)$ pair have been investigated. Figure 4 (b) and Figure 4 (c) show that using the $L^+(r_1, r_2)/L^-(r_1, r_2)$ loss boosted/penalized interactions of the $[r_1 n, r_2 n]$-th orders.

Second, in spite of that, we conducted new experiments to further investigate the effects of multiple $(r_1, r_2)$ pairs. We used the following five sets of experimental settings, including (i) $L^+(0, 0.3)$

Table 3: Comparison of the running time between normally training DNNs by 200 epochs and training DNNs by 200 epochs with the proposed losses.

|  | CIFAR-10 | | Tiny-ImageNet | |
| --- | --- | --- | --- | --- |
| Model | AlexNet | VGG16 | AlexNet | VGG16 |
| Normally training DNNs | 0.46 h | 0.83 h | 0.89 h | 9.25 h |
| Training using $L^+(r_1, r_2)$ | 1.36 h | 2.35 h | 1.87 h | 27.28 h |
| Training using $L^-(r_1, r_2)$ | 1.27 h | 2.12 h | 1.82 h | 26.68 h |

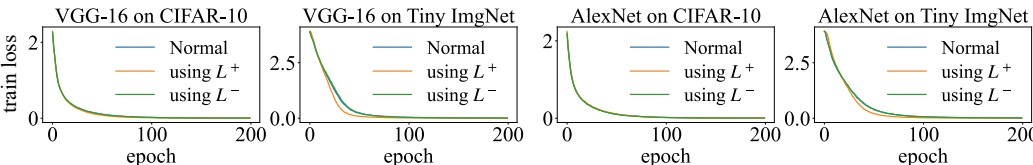

Figure 19: Curves of classification loss $\text{Loss}_{\text{classification}}$ *w.r.t.* training epochs for normally training DNNs and training DNNs with the proposed losses.

and $L^+(0.7, 1.0)$; (ii) $L^+(0.3, 0.7)$ and $L^+(0.7, 1.0)$; (iii) $L^+(0.2, 0.4)$ and $L^+(0.6, 0.8)$; (iv) $L^-(0, 0.5)$ and $L^+(0.6, 1)$; (v) $L^+(0, 0.5)$ and $L^-(0.6, 1)$. We used these parameters to train DNNs on the CIFAR-10 dataset. Figure 16(a) shows that $L^+(r_1, r_2)$ and $L^+(r'_1, r'_2)$ loss simultaneously boosted interactions of the $[r_1 n, r_2 n]$-th orders and the $[r'_1 n, r'_2 n]$-th orders. For example, by using $L^+(0, 0.3)$ and $L^+(0.7, 1.0)$, the trained DNN simultaneously boosted both interactions of the $[0, 0.3n]$-th orders and interactions of the $[0.7n, n]$-th orders.

In addition, we conducted experiments on two tabular datasets, *i.e.*, the census dataset and the commercial dataset. We extended above five different experimental settings (*w.r.t.* $r_1$ and $r_2$) for image data to the training of DNNs in tabular data. Thus, we trained five MLP-5 networks on each tabular dataset following each experimental setting. The architecture of the MLP-5 has been introduced in Section 3.1. Figure 17 shows that we could use $L^+(r_1, r_2)$ and $L^+(r'_1, r'_2)$ loss to simultaneously boost interactions of the $[r_1 n, r_2 n]$-th orders and interactions of the $[r'_1 n, r'_2 n]$-th orders.

Third, we conducted new experiments to investigate the effects of the parameter $\lambda$. Specifically, we fixed $\lambda_1 = 0$, and adjusted $\lambda_2 = 0.01, 0.1, 1, 5$ to study the effects of $\lambda_2$. We trained DNNs on the Tiny-ImageNet dataset. First, we trained high-order DNNs by penalizing interactions of the $[0, 0.5n]$-th orders. Figure 16(b) shows that the interaction strength of the $[0, 0.5n]$-th orders decreased as $\lambda_2$ increased. The results indicated that the larger $\lambda_2$ value penalized interactions of the $[0, 0.5n]$-th orders more significantly. Second, we trained low-order DNNs by penalizing interactions of the $[0.7n, n]$-th orders. Similarly, we found that the strength of these high-order interactions also decreased as $\lambda_2$ increased. This further indicated that the larger $\lambda_2$ value penalized the high-order interactions more significantly. We also conducted experiments on the census dataset and the commercial dataset. Specifically, we fixed $\lambda_1 = 0$, and adjusted $\lambda_2 = 0.1, 1, 10$. We trained high-order DNNs by penalizing interactions of the $[0, 0.5n]$-th orders. Here, we used the MLP-5 networks mentioned in Section 3.1. Figure 18(a) also shows the same pattern that the larger $\lambda_2$ value penalized interactions of the $[0, 0.5n]$-th orders more significantly. These results showed that the parameter $\lambda_2$ controlled the strength of penalization.

Furthermore, we investigated the effects of $\lambda_1$ on two tabular datasets. Specifically, we fixed $\lambda_2 = 0$, and adjusted $\lambda_1 = 0.1, 1, 10$ to study the effects of $\lambda_1$. We trained middle-order DNNs by boosting interactions of the $[0.3n, 0.7n]$-th orders. Here, we used the MLP-5 networks mentioned in Section 3.1. Figure 18(b) shows that the parameter $\lambda_1$ controlled the strength of boosting interactions.

### C.2.5 TIME COMPLEXITY ANALYSIS.

We compared the running time between training DNNs by 200 epochs using the proposed losses and normally training DNNs by 200 epochs. We trained these models with a mini-batch size of 128 on a single NVIDIA GeForce RTX 3090 GPU and used 4 subprocesses in data loading. Table 3 shows that the time cost of training DNNs with the proposed losses was about three times as much as the time cost of normally training DNNs, which was acceptable in real applications.

Figure 20: The distributions of interaction strength $\bar{J}^{(m)}$ of different DNNs using new metric. These DNNs were trained on various image datasets and tabular datasets.

Table 4: Mean values and standard deviations ($\mu \pm \sigma$) of the projections at the first five principal directions of the gradient(at initialization), which were computed under different sizes of contexts.

|  | $|S| = 0.05n$ | $|S| = 0.5n$ | $|S| = 0.95n$ |
|---|---|---|---|
| The first direction | -0.0024±0.1004 | 0.0025±0.1740 | 0.0103±0.2277 |
| The second direction | 0.0089±0.0890 | 0.0003±0.1434 | 0.0076±0.1906 |
| The third direction | -0.0005±0.0855 | -0.0012±0.1350 | 0.0139±0.1819 |
| The fourth direction | -0.0029±0.0802 | -0.0012±0.1250 | -0.0010±0.1543 |
| The fifth direction | 0.0000±0.0766 | 0.0005±0.1161 | 0.0022±0.1453 |

In fact, a more reasonable experiment should compare the time of training the DNN to convergence, *i.e.* the time cost of training a DNN with a certain decrease of the classification loss. To this end, we plotted curves of the classification losses *w.r.t.* training epochs of normally trained DNNs and DNNs trained with the proposed $L^+(r_1, r_2)$ and $L^-(r_1, r_2)$ losses, respectively. Figure 19 shows that the three types of DNNs all achieved convergence at the 200-th epoch, which validated the reliability of the above comparison on the time cost.

### C.2.6 OTHER VALIDATION EXPERIMENTS

**Using new metric to validate bottleneck.** Besides, to further remove effects of magnitudes of $|I^{(m)}(i, j|x)|$ on different samples for fair comparison, we adopted the following new metric.

$$\bar{J}^{(m)} = \mathbb{E}_{x \in \Omega} \frac{\mathbb{E}_{i,j}[|I^{(m)}(i, j|x)|]}{\mathbb{E}_{m'}[\mathbb{E}_{i,j}[|I^{(m')}(i, j|x)|]]} \tag{13}$$

where $\Omega$ denotes the set of all samples. We found that using the new metric resulted in few changes to the distribution of the interaction strength. Figure 20 shows a similar representation bottleneck phenomenon when we used the new metric, *i.e.*, the middle-order interaction was less likely to be encoded in DNNs.

**Validation of zero-mean assumption in Theorem 1.** We further verified the reliability of the zero-mean assumption in Theorem 1. To this end, we analyzed the mean value $\mathbb{E}_{i,j,S}[\frac{\partial \Delta v(i,j,S)}{\partial W}]$ of the gradient *w.r.t.* the parameter $W$ at the first convolutional layer when we trained ResNet-18 on the Tiny-ImageNet dataset.

Since the above gradient $\frac{\partial \Delta v(i,j,S)}{\partial W}$ is a high-dimensional vector, it is difficult to visualize the expectation of all dimensions. Alternatively, we inferred and analyzed the projections on the first five principal directions of the gradient. Then, the validation was simplified as analyzing the mean value and standard deviation of each projection.

Specifically, we computed five principal directions as $o_1, o_2, \ldots, o_5$. We examined the mean value and the standard deviation of the gradient strength projected on each direction $o_i$, over gradients *w.r.t.* different triplets $(i, j, S)$ computed on 10 different samples. Given each input sample, we computed gradients $\frac{\partial \Delta v(i,j,S)}{\partial W}$ for 10 pairs of $(i, j)$ and 1000 contexts $S$.

The examination was conducted when $|S| = 0.05n, |S| = 0.5n$, and $|S| = 0.95n$. Tables 4 and 5 report the above mean values and standard deviations for the network at initialization and the network trained for 40 epochs, respectively. Tables 4 and 5 show that these mean values were almost 0, which validated the zero-mean assumption in Theorem 1.

In addition, for simplicity, we used $\sigma^2$ to denote the variance of the gradient $\frac{\partial \Delta v(i,j,S)}{\partial W}$ in Theorem 1, without distinguishing different sizes of contexts $S$. Tables 4 and 5 show that there existed

Table 5: Mean values and standard deviations ($\mu \pm \sigma$) of the projections at the first five principal directions of the gradient (at 40 epochs), which were computed under different sizes of contexts $S$.

|  | $\lvert S \rvert = 0.05n$ | $\lvert S \rvert = 0.5n$ | $\lvert S \rvert = 0.95n$ |
|---|---|---|---|
| The first direction | -0.0015±0.4037 | 0.0003±0.8675 | 0.0423±1.5470 |
| The second direction | -0.0209±0.3042 | 0.0042±0.5653 | 0.0802±0.9754 |
| The third direction | -0.0284±0.2817 | 0.0070±0.5164 | 0.0269±0.7845 |
| The fourth direction | 0.0208±0.2315 | 0.0003±0.5057 | 0.0281±0.7231 |
| The fifth direction | -0.0018±0.1853 | 0.0082±0.4178 | -0.0358±0.6715 |

some small difference between variances of the gradient when we considered different sizes of contexts $S$. However, such difference is negligible in the proof and does not affect the conclusion of Theorem 1. This is because the learning strength of the $m$-th order interaction is proportional to $\frac{n-m-1}{n(n-1)} / \sqrt{\binom{n-2}{m}}$, whose difference between different sizes of contexts was much larger than the above difference in variance.

### C.2.7 CLARIFICATION OF THE INSIGHT OF TABLE 1

Table 1 compared the classification accuracy (left part) and compared adversarial robustness (right part) between DNNs encoding different orders of interactions.

Table 1 did provide some new insights. By comparing results in Table 1 (left) with results in Table 1 (right) and Figures 6, 9, 13, and 14, we found that although the interactions of different orders performed differently in both adversarial robustness (Table 1 (right)) and the capacity of encoding structural information (Figures 6, 9, 13, and 14), interactions of different orders had similar classification accuracies. It showed that although interactions of different orders had their distinctive properties, such difference did not necessarily affect their classification accuracies.

