# OpenReview forum: "DISCOVERING AND EXPLAINING THE REPRESENTATION BOTTLENECK OF DNNS"
_ICLR.cc/2022/Conference — ICLR 2022 Oral_

### Official Review · Reviewer_FMrV · 2021-10-31

**Correctness:** 3
**Technical Novelty And Significance:** 3
**Empirical Novelty And Significance:** 3
**Recommendation:** 8
**Confidence:** 4

**Main Review:**

Strengths:

1) The novelty of the paper lies in using the multi-order interaction proposed in Zhang et al. (2020) to understand the complexity of interactions in DNNs.

2) The major strength of the paper lies in the extensive study of different neural networks and datasets to back the claim. Also, the effect of the proposed loss functions $L^{+}$ and $L^{-}$ on deeper networks like AlexNet, VGG-16, and ResNet-18/20 is quite interesting.


Suggestions:

1) The number of sets $S$ s.t. $|S| = m$ is given by $n \choose m$. This function increases from $m = 1 \ldots \frac{n}{2}$ and decreases thereafter till $n$. Thus, for achieving large $I^{(m)}(i, j)$ at $m = \frac{n}{2}$, the model needs to capture dependence between the variables $i$ and $j$ for significantly higher number of contexts, compared to the cases when $m=3$ (local context) or $m=n-2$ (global context). One may not need the dependence to be large for all possible contexts in the intermediate context length case.  Hence, can the authors give an example to explain why the increasing number of contexts in the intermediate contextual interactions doesn't necessarily lead to a small $I^{(m)}(i, j)$ always?

2) Since, the authors consider $v(S|x) = \log \frac{1 - \Pr[\hat{y} = y^{\star}]}{  \Pr[\hat{y} = y^{\star} }$ for computing $J^{(m)}$ (which depends on the probability scores of a data example), I believe the magnitudes of $v(S|x)$ aren't necessarily comparable for different examples. Then, isn't the following definition of $J^{(m)}$ a proper relative interaction strength to measure?
\begin{equation*}
    J^{(m)} = \mathbb{E}_{x \in \Omega} \frac{ \mathbb{ E }   I^{(m)} (i, j \mid x)   }{\mathbb{E_k } \mathbb{E} |I^{(k)} (i, j | x) |
 }
\end{equation*}

The above interaction measure measures the strength of the contextual utility for each example.

3) In the adversarial robustness experiments, are the low-order and intermediate-order interactions more robust to adversarial attacks? It will be great to have some experiments for intermediate-order interactions, to possibly showcase the fact that neural networks are susceptible to adversarial attacks because of the cognition gap.






**Summary Of The Paper:**

The authors study the complexity of interactions between input variables in deep neural networks (DNNs). They use a multi-order interaction utility between pairs of variables to represent the interaction complexity between input variables encoded in DNNs. The results can be summarized as follows:

1) DNNs are more likely to encode simple and complex interactions between pairs of input variables but aren't good enough to encode the intermediate ones. The authors define the simplicity of interaction depending on its contextual complexity.

2) The authors develop novel loss functions to encourage/penalize the interactions of specific complexities and then, investigate the representation capacities of the learned DNNs.



**Summary Of The Review:**

Overall, the paper makes an important evaluation study to further the understanding of the representation capacity of DNNs. The novelty is in using the multi-order interaction proposed in Zhang et al. (2020) to understand the complexity of interactions in DNNs and the proposal of different loss functions to encode/penalize interactions of specific complexities. Hence, I am leaning towards a positive score for the paper.

---

### Official Review · Reviewer_4jZH · 2021-11-02

**Correctness:** 4
**Technical Novelty And Significance:** 4
**Empirical Novelty And Significance:** 4
**Recommendation:** 8
**Confidence:** 3

**Main Review:**

The research topic of this paper is significant in understanding the representation power of deep neural networks, and is interesting to the entire community of deep learning. The writing of this paper is clear enough, and the main contents are easy to follow.

Strengths:
- The definition of the relative interaction strength is elegant.
- The theoretical and numerical analysis in Section 3.2 is novel and interesting.
- The loss functions to encourage or penalize interactions of certain orders are also novel and very natural.
- There are experiments showing the effects of these loss functions and connecting them to properties such as accuracy and robustness.

However, there are several concerns of this paper. The following are about theory.
- Regarding Thm 1, how reasonable is the assumption that $\frac{\partial}{\partial W} \Delta v(i,j,S)$ is a Gaussian? At least some numerical validations should be made here.
- Regarding the simulation in Fig 3, there is still a tiny gap between the blue and orange curves, which does not look like a random pattern. Can you explain where this gap comes from?
- Regarding equation 5, there are two issues. First, it is confusing to reload the notation $\Delta v$. I would recommend using another symbol. Second, are you missing an expectation in this equation?
- In equation 7, it is clearer to write $L^+(r_1,r_2)$ rather than $L^+$. Simimlar in equation 8.
- Regarding equation 9 and simulations in Fig 4, it is possible to use multiple $L^+$ or $L^-$ with different $(r_1,r_2)$ pairs? If so, why are you not using it? I feel this can provide more flexibility and is therefore potentially more useful.

There are also concerns regarding experiments. In summary, the experiments are not extensive and cannot well support the findings.
- Regarding accuracy, what can we read from Table 1 (left)? The numbers seem random.
- Regarding structural representations, Fig 5 & 6 make some sense, but what about other masking methods?
- Regarding robustness, Table 1 (right) tells us that penalizing low-order and boosting high-order interactions harm robustness. What about doing the opposite, i.e. penalizing high-order and boosting low-order interactions? Does it improve robustness?
- In general, the experiments are not extensive. In order to better understand the effects of the proposed loss functions, there should be experiments carefully adjusting the $\lambda$'s and $(r_1,r_2)$'s so that we can extract general patterns of the effects.
- In addition, I do not see any significant improvement by using the proposed loss functions. Are there scenarios where using the proposed loss functions outperforms standard training in any of the metric (accuracy, robustness)?
- Finally, figures and tables are a bit small so hard to read.

Scores can be modified based on authors' feedback and additional experiments.

----------------------
After rebuttal: I have carefully read the response and added experiments.

- The revised theory with milder assumption is great.
- The additional experiments are excellent and provide a much more complete picture of the proposed theory.

I have modified the score accordingly.

**Summary Of The Paper:**

The paper looks at representation bottleneck of deep neural networks, from the multi-order interaction (within subsets of pixels) point of view. Specifically, the paper provides a novel definition on the relative interaction strength, and then analyzes this quantity both theoretically and numerically. The paper next proposes two loss functions that encourage or penalyze interactions of certain orders, and conducts multiple experiments on different combinations of these loss functions.

**Summary Of The Review:**

Theory is excellent but there is space for improvement in the experiments.

----------------------
After rebuttal: experiments are extensive and insightful enough.

---

### Official Review · Reviewer_t6sY · 2021-11-02

**Correctness:** 3
**Technical Novelty And Significance:** 4
**Empirical Novelty And Significance:** 3
**Recommendation:** 10
**Confidence:** 4

**Main Review:**

[Strength]
1. This paper discovers a very interesting tendency (called representation bottleneck) in existing DNNs, i.e., a DNN is usually more likely to encode low-order and high-order interactions, but difficult to encode middle-order interactions. $\\\\$
2. This paper analyzes theoretically the reason behind the bottleneck. Moreover, the simulated results based on Theorem 1 well match the real cases, which validates the theoretical reason. $\\\\$
3. From the perspective of the bottleneck, the authors explore the difference between DNN learning and human learning, which provides the key idea of the proposed two losses. $\\\\$
4. To relieve the bottleneck phenomenon, the authors propose two novel losses to enable DNNs to learn interactions of any specific orders. $\\\\$
5. Experimental results demonstrate the effectiveness of the two losses, and show that middle-order interactions can provide useful information for classification. $\\\\$

[Weakness]
1. The explanation for the representation bottleneck is not intuitive enough.  $\\\\$
2. There is a lack of discussion why the trained DNNs do not achieve a significant improvement than a normally trained DNN.$\\\\$
3. The experiments on adversarial robustness can not well support the conclusions, since they were only conducted on tabular datasets. $\\\\$

[Suggestions]
1. I would suggest the authors give an additional illustration of the intuition behind the proof for the bottleneck. $\\\\$
2. It’s expected to give guidance on how we can use the two losses to boost the classification performance. $\\\\$
3. It’s not clear why the output change in Eq.(5) can represent interactions of specific orders. $\\\\$
4. The authors are suggested to report results in terms of adversarial accuracy on more datasets such as image data.  $\\\\$
5. The authors are suggested to clarify whether the proposed two losses will increase the time complexity.  $\\\\$

***
After carefully reading the response, other reviews, and the revised version of the manuscript.

1. The added proof skeleton makes the theory more clear and intuitive.
2. New experimental results show a strong connection between representation capacity and interaction orders, which may inspire follow-up studies.

Overall, I think this work is insightful and may make a great impact on both training and explaining of deep neural networks. I have modified the score accordingly.





**Summary Of The Paper:**

This paper studies the representation ability of DNNs from the perspective of interactions (the multi-order interaction). The authors discovered an interesting representation bottleneck phenomenon, i.e., in a normally trained DNN, low-order and high-order interaction patterns are easy to be learned, while middle-order interaction patterns are difficult to be learned. Moreover, they give a theoretical explanation that the bottleneck origins from that different interaction patterns have different learning strengths (gradients). To relieve the bottleneck problem, the authors propose two novel loss functions, which allow the model to encode interactions of specific orders, including middle-order interactions. The experiments have validated the effectiveness of the proposed losses. Finally, the authors explored some properties of DNNs encoding different-order interactions.

**Summary Of The Review:**

This paper discovers and theoretically explains an interesting representation bottleneck in existing DNNs, and relieves the bottleneck by proposing two novel losses. As the discovered bottleneck reflects the common difficulty to learn middle-order interactions, this work may provide a new direction to train DNNs.

---

### Official Review · Reviewer_cEoR · 2021-11-02

**Correctness:** 3
**Technical Novelty And Significance:** 3
**Empirical Novelty And Significance:** 3
**Recommendation:** 8
**Confidence:** 4

**Main Review:**

Strengths
- The paper is well-written and easy to follow. In particular, Figure 1 and 2 support the concept of representation bottleneck reasonably well.
- Two questions the authors raise about DNNs are rational, and prove convincingly that this phenomenon is a common problem of DNNs.
- The idea of describing the representation of the DNN using multi-order interaction utility is good and easy to understand.

Weaknesses
- The authors propose two losses to encourage/penalize DNNs to learn interactions of specific orders and show the results in Figure 4. This loss can stimulate interactions of specific orders well. High-order DNN seems to have been fully explained and experimented, but not on middle-order and low-order interaction. I wonder why the authors didn't explain it in detail.
- The authors conduct several experiments, but these show similar results, making it difficult to gain new insights. And it is difficult to understand what Table 1 means and shows.
- Although this paper shows good enough insight to others, it would be good to present directions for solving the problem of representation bottleneck or providing a clear problem definition for future works.
- I know it’s due to the limitations of the amount, but putting too many formulas and terms into the sentence. Instead of including all the formulas in the sentence, it would be better to reorganize it and make it easier to legible.


**Summary Of The Paper:**

This paper discovers and theoretically proves the representation bottleneck phenomenon that indicates cognition gap between DNNs and humans. This paper proposes losses to encourage or penalize the DNN to control interactions of specific complexities, and analyze the representation capacities of interactions.

**Summary Of The Review:**

This paper introduces and proves the representation bottleneck, which is a common  phenomenon in the DNNs. This is well-written and easy to understand. There seems to be some shortcomings in the experiments and conclusion, but the paper provides good insights to others.

---

### Decision · Program_Chairs · 2022-01-20

**Decision:**

Accept (Oral)

**Comment:**

This paper makes an important evaluation study to further the understanding of the representation capacity of DNNs. The novelty is in using the multi-order interaction proposed in Zhang et al. (2020) to understand the complexity of interactions in DNNs. The authors discovered an interesting representation bottleneck phenomenon, i.e., in a normally trained DNN, low-order and high-order interaction patterns are easy to be learned, while middle-order interaction patterns are difficult to be learned. They also propose two novel loss functions, which allow the model to encode interactions of specific orders, including middle-order interaction. All reviews are positive.